# Differential stability and dynamics of DNA-based and RNA-based coacervates affect non-enzymatic RNA chemistry

Karina K. Nakashima [1,2,8], Fatma Zohra Mihoubi[1,2,8], Jagandeep S. Saraya [3], Kieran O. Russell [4], Fidan Rahmatova [1,2], James D. Robinson [3], Maria Julia Maristany [4,5], Jan Huertas [4,6], Roger Rubio-Sánchez [7], Rosana Collepardo-Guevara [4,5,6] ✉, Derek K. O'Flaherty [3] ✉ & Claudia Bonfio [1,2] ✉

The RNA-peptide world hypothesis postulates the early co-evolution of RNA and peptides that led to the emergence of non-enzymatic RNA replication and peptide synthesis. Although nucleotides and amino acids have been shown to form and polymerise under prebiotic conditions, the origins of their synergy remain unclear. We propose that cooperation between DNA, RNA and peptides could have stemmed from their co-localisation in early biological compartments. Here, we show that heterogeneous mixtures of prebiotic oligonucleotides and peptides can spontaneously assemble into primitive coacervates. Experimental and computational studies reveal that peptide/nucleic acid coacervates are highly robust and form under a notably broad range of conditions. RNA-based coacervates are exceptionally stable and, in the presence of DNA, very fluid, which facilitates diffusion of reactive oligonucleotides and supports prebiotic RNA chemistry. Our findings suggest that coacervation may have occurred very early on the evolutionary timeline and fostered the emergence of a nucleic acid-peptide world. This study provides insights into the prebiotic role of coacervates and reconsiders their significance for the origins of life and the emergence of primitive replication and translation systems.

The RNA-peptide world hypothesis proposes the early co-evolution of RNAs and peptides, from which RNA replication and peptide synthesis may have emerged[1,2]. It was recently shown that RNA nucleotides and amino acids form non-enzymatically, alongside DNA nucleotides, in prebiotic conditions[3–5] and polymerise into short RNA and DNA oligomers, and peptides[6–8]. Although no defined prebiotic role has been proposed for DNA until its genetic takeover of RNA[9], canonical and non-canonical RNAs were reported to template RNA and DNA polymerisation[10] and direct peptide synthesis[1,2,11], and short peptides derived from the ribosomal core enhanced ribozyme activity[12]. Yet, how the primordial synergy between nucleic acids and peptides originated remains unknown. An intriguing hypothesis relies on the ability of the building blocks of life to cooperate upon co-localisation by means of compartmentalisation early on the evolutionary timeline[13].

Biomolecular condensates, generated through the liquid-liquid phase separation of RNA and proteins, have been proposed as vestiges

[1]Department of Biochemistry, University of Cambridge, Cambridge, UK. [2]Institut de Science et d'Ingénierie Supramoléculaires, CNRS UMR 7006, University of Strasbourg, Strasbourg, France. [3]Department of Chemistry, University of Guelph, Guelph, ON, Canada. [4]Yusuf Hamied Department of Chemistry, University of Cambridge, Cambridge, UK. [5]Cavendish Laboratory, Department of Physics, University of Cambridge, Cambridge, UK. [6]Department of Genetics, University of Cambridge, Cambridge, UK. [7]Department of Chemical Engineering and Biotechnology, University of Cambridge, Cambridge, UK. [8]These authors contributed equally: Karina K. Nakashima, Fatma Zohra Mihoubi. ✉e-mail: rc597@cam.ac.uk; doflaher@uoguelph.ca; cb2036@cam.ac.uk

of primitive cells[14] because of their ability to spatially regulate cellular biochemistry[15]. Employed as in vitro models of biomolecular condensates, complex coacervates comprising peptides and functional oligonucleotides, e.g., ribozymes, were shown to take up dilute solutes, enable diffusion within and between the compartment and its environment, and host prebiotic reactions, e.g., ribozymatic activity[14,16–21]. Interestingly, it was recently reported that coacervates comprising heteropeptides with low charge density enhance ribozyme mobility and maximise $Mg^{2+}$ uptake compared to coacervates composed of polyarginines[17]. Complex coacervation results from electrostatic interactions between oppositely charged polymers, such as positively charged polyarginines or polylysines, and negatively charged inorganic polyphosphates, polyglutamates, polyaspartates, nucleotide polyphosphates or sequence-specific nucleic acids[14,19,22,23]. However, in any prebiotic scenario, non-coded polymerisation pathways would have likely afforded DNA and RNA oligomers of limited length and high compositional heterogeneity, besides complex mixtures of peptides[6,24,25]. As such, the prebiotic feasibility of peptide/oligonucleotides coacervates, *i.e.*, whether they would have spontaneously emerged from simple, prebiotic molecules or relied upon the synthesis of long, coded, functional polymers (i.e., homopeptides and ribozymes), is yet to be understood.

Here we show that prebiotically plausible heterogeneous oligonucleotides form coacervates even with tri- and dipeptides. Through experimental and computational studies, our findings indicate that coacervation could have occurred early in the evolutionary timeline, possibly simultaneously with the emergence of a nucleic acid-peptide world. Focusing on a prebiotic context, we systematically compare peptide/peptide and peptide/nucleic acid coacervates and demonstrate that the latter form under a much broader range of conditions; and that RNA-based coacervates are remarkably more stable than both peptide/DNA and peptide/peptide analogues. Importantly, we find that DNA minimally affects the stability of RNA-based coacervates but critically enhances the diffusion of reactive oligonucleotides involved in non-enzymatic RNA polymerisation. Our results suggest that DNA played an early role in compartmentalisation to enable the emergence of primitive coacervates capable of hosting RNA biochemistry. Our work reconsiders the significance of primitive coacervates to support replication and translation in a general compartmentalised nucleic acid-peptide world to further our understanding of the possible origins of life.

## Results

### RNA coacervates are highly stable

Arginine (Arg, R) homopeptides were recently shown to undergo liquid-liquid phase separation in the presence of negatively charged molecules of low multivalency, e.g., nucleotide phosphates and glutamic acid (Glu, E) or aspartic acid (Asp, D) homopeptides[17,19,26]. Ribozymes or long sequence-specific nucleic acids (NAs) also were reported to undergo phase separation with positively charged ions, polyamines and peptides[16–18,27–29]. However, prebiotic polymerisation processes would have mainly produced short, non-functional oligonucleotides and peptides, for which the *coacervating* propensity is unknown. In view of the prebiotic plausibility of both ribonucleotides and deoxyribonucleotides[5,30,31], we investigated the propensity of single-stranded (ss) DNA and RNA oligomers for coacervation with short Arg peptides. Evidence suggests DNA would have been present in an RNA world[32–35], but its role remains unclear[32] until the genetic takeover of RNA by DNA as information carrier[9].

We first assessed the salt stability of coacervates made of Arg tetramers ($R_4$) with $DNA_8$ (($ACTG)_2$) or $RNA_8$ (($ACUG)_2$) and compared it with that of previously studied[19] coacervates comprising negatively charged peptides (Glu decamers, $E_{10}$) (Supplementary Fig. S1 and Supplementary Tables S1 and S2). For this study, we used short Arg peptides as *model* peptides given the prebiotic availability of arginine[3]

and the expected length of peptides synthesised under prebiotic conditions[24]. Additionally, we chose oligonucleotide sequences with an ACTG/ACUG motif to avoid the formation of secondary structures and minimise nucleobase/sequence biases. Turbidity measurements upon titration of NaCl allowed us to determine the critical salt concentration (CSC) of peptide/oligonucleotides and peptide/peptide mixtures (Supplementary Fig. S2 and Supplementary Table S3). CSC is conventionally taken as an indication of coacervate robustness[19] and defined as the highest NaCl concentration tolerated before complete dissolution of the droplets. CSC values at different [Arg]:[nucleotide] ratios were plotted to delineate the phase diagram of peptide/peptide and peptide/nucleic acid mixtures (Fig. 1a and Supplementary Figs. S3–S5 and Supplementary Table S4).

In line with previous observations[19], we found that $R_4$/$E_{10}$ mixtures do not form coacervates. When a longer positively charged peptide ($R_{10}$) was used with $E_{10}$, the maximum salt stability of the resulting coacervates was obtained when the two peptides were present in equimolar charge concentrations. A re-entrant transition was observed with excess $R_{10}$, which suggests that peptide/peptide mixtures form coacervates only when the charge concentration mismatch is minimal (Fig. 1a). When $E_{10}$ was replaced by $DNA_8$ and $RNA_8$, we observed coacervation across a broader range of environmental conditions (Supplementary Figs. S6 and S7). Upon increasing the concentration of arginine while keeping fixed that of oligonucleotide, salt stability curves plateaued at 4:1 [Arg]:[nucleotide]. No re-entrant transition was observed even with a high polymer charge mismatch, thus generating wider phase co-existence regions than those of peptide/peptide coacervates (Fig. 1a). The propensity of DNA oligonucleotides to undergo phase separation with peptides in mismatched charge concentrations suggests that the peptide/DNA coacervates may have been more likely to occur in a prebiotic setting than their peptide/peptide analogues.

Surprisingly, the salt tolerance of $R_4$/$RNA_8$ is 2.2 times higher relative to that of the $R_4$/$DNA_8$ mixture, rising from 99.3 mM to 215.9 mM NaCl at 4:1 [Arg]:[nucleotide]. Given the high CSC values of peptide/nucleic acid coacervates at 4:1 [Arg]:[nucleotide], we performed all the following experiments at this charge ratio, unless otherwise specified. A greater tendency of RNA oligomers to form coacervates over their DNA counterparts was also confirmed by measuring the minimal concentration of oligonucleotide and peptide required for coacervation, which is 2-fold lower for $R_4$/$RNA_8$ mixtures relative to $R_4$/$DNA_8$ mixtures (Fig. 1b). Intrigued by the enhanced salt stability of peptide/RNA coacervates, we used hot-stage epifluorescence microscopy[36] to evaluate their temperature susceptibility relative to analogous peptide/DNA and peptide/peptide coacervates (Fig. 1c and Supplementary Fig. S8).

Along the heating ramp, full dissolution of the $R_4$/$DNA_8$ coacervates was observed at ≈45 °C. Conversely, $R_4$/$RNA_8$ coacervates showcase greater thermal stability, dissolving only at ≈60 °C. A similar thermal stability was observed for peptide/peptide coacervates only when longer polymers were employed ($R_{10}$/$E_{10}$) (Supplementary Fig. S8). In all cases, cooling led to coacervation, which confirmed the reversibility of the assembly process. Although an additional hydroxyl group was shown to increase the CSC of coacervates comprising small metabolites[37], the unprecedented difference in the salt and thermal stability of DNA and RNA coacervates suggests stronger interactions between RNA and peptides than between DNA and peptides.

The differences in thermal and salt stability observed for $R_4$/$DNA_8$ and $R_4$/$RNA_8$ coacervates suggest distinct peptide length requirements for coacervation when RNA oligomers are used instead of DNA analogues. We found that $RNA_8$, but not $DNA_8$, forms coacervates with Arg trimers ($R_3$) (CSC = 54.2 mM) (Supplementary Table S3); still, four extra nucleobases ($DNA_{12}$) enable coacervation with $R_3$. Similarly, droplets were observed when Arg dimers ($R_2$) were mixed with $RNA_{20}$ but not with DNA oligonucleotides up to 50 nucleotide-long

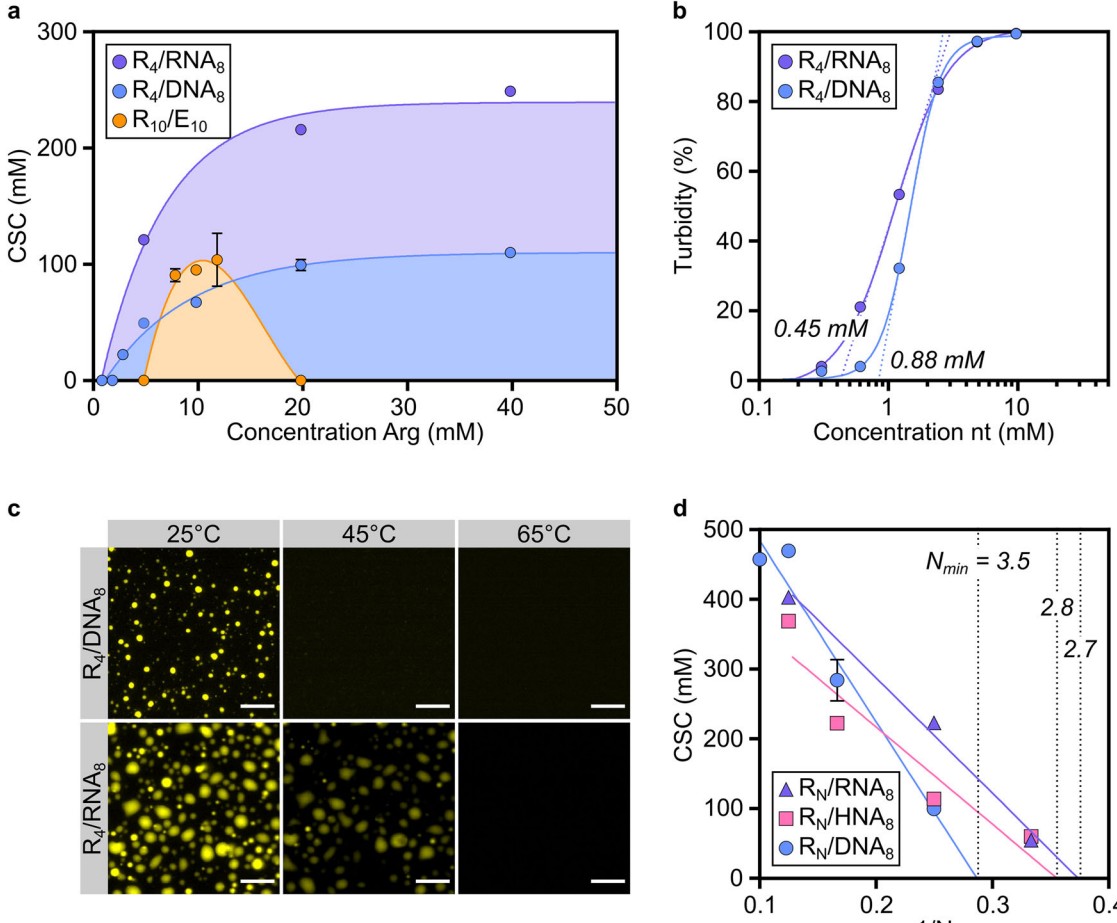

**Fig. 1 | Peptide/RNA coacervates exhibit higher robustness than peptide/DNA coacervates. a** Salt stability of peptide/peptide and oligonucleotide/peptide coacervates. Critical salt concentrations (CSCs) were measured through turbidity measurements of peptide/peptide and oligonucleotide/peptide solutions by titration with NaCl in 25 mM HEPES, pH 7.5 and at room temperature. In all experiments, the anion concentration was kept constant ([nt] = 5 mM and [glutamic acid] = 10 mM). **b** Turbidity curves for $R_4/DNA_8$ and $R_4/RNA_8$ as a function of nucleotide concentration. The dotted lines are tangents to the inflection point, used to determine the minimal concentration required for coacervation (indicated in the graph). **c** Thermal stability of $R_4$ coacervates with $DNA_8$ and $RNA_8$. 1% $Cy_3$-$(TGAC)_2$ was used for visualization. Scale bars are 10 μm. **d** Estimation of the minimal peptide

length ($N_{min}$) required for coacervation for a given nucleic acid composition. $R^2$ values for the linear fit: 0.96 ($DNA_8$), 0.89 ($HNA_8$) and 0.99 ($RNA_8$). R = Arginine, E = Glutamic acid, nt = nucleotide, $DNA_8$ = 8-deoxyribonucleotide-long mixed-sequence oligonucleotide (($ACTG)_2$), $RNA_8$ = 8-ribonucleotide-long mixed-sequence oligonucleotide (($ACUG)_2$), $HNA_8$ = 8-nucleotide-long mixed-sequence oligonucleotide comprising deoxyribonucleotides and ribonucleotides (ArCrU-GArCrUG). $n$ = 3 for data in (**a**); mean and SD values are provided. CSC values in (**a**) can be found in Supplementary Table S4; values in (**d**) can be found in Supplementary Table S3. Parameters from the linear fit in (**d**) are in Supplementary Table S6. Source data are provided as a Source Data file.

(Supplementary Table S5 and Supplementary Fig. S9). To our knowledge, $R_2$ is the shortest peptide reported to be engaged in complex coacervation.

We next computed the minimal peptide length ($N_{min}$) required for coacervation with any given oligonucleotide to delineate the precise co-existence boundaries in the related phase space (Supplementary Table S6). We determined the CSC for coacervates made of a series of oligonucleotides ($DNA_8$, $RNA_8$, $DNA_{12}$ and $RNA_{12}$) with Arg peptides of various lengths ($R_N$) (Fig. 1d and Supplementary Fig. S10). In agreement with predictions for long polymers[38], we observed a linear relationship between the CSC and the inverse of polymer length (1/N) for primitive coacervates. We confirmed that at least an Arg tetramer is required to form coacervates with $DNA_8$ ($N_{min}$ = 3.5), whereas coacervation occurs with $RNA_8$ and the shorter $R_3$ ($N_{min}$ = 2.7) (Supplementary Table S6). Because chimeric RNA-DNA oligonucleotides would have likely emerged from a prebiotic pool of ribonucleotides and deoxyribonucleotides[30,31,39], we also tested an oligonucleotide comprising 50% RNA and 50% DNA nucleotides ($HNA_8$) and observed an $N_{min}$ value similar to that obtained for $RNA_8$ ($N_{min}$ = 2.8) (Fig. 1d). An

analogous trend was observed for longer oligonucleotides, with $RNA_{12}$, hybrid strands ($HNA_{12}$) and mixed DNA-RNA oligomers predicted to form coacervates with Arg trimers (Supplementary Fig. S11). These results suggest that the effect of ribonucleotides or RNA oligomers in a heterogeneous mixture with Arg peptides could have overcome that of deoxyribonucleotides and DNA oligomers and led to the emergence of coacervates with minimal length requirements and salt stability similar to those of a pure peptide/RNA system.

Homopolymeric DNA and RNA sequences have been widely studied for their ability to form coacervate models[38,40]. However, purines are only slightly more reactive than pyrimidines in template-free non-enzymatic RNA polymerisation[41], so heteropolymeric sequences would have likely been more abundant than homopolymeric analogues on early Earth. We thus investigated how oligonucleotide sequence and charge influences coacervation. Polycytosine and polyguanine decamers formed solid-like aggregates; but polyadenine and polythymine decamers formed coacervates with substantially lower CSCs than those made of heteropolymeric DNA sequences comprising all four nucleotides (Supplementary Fig. S12). Similarly, the minimal

oligonucleotide length required for coacervation with $R_6$ is almost 2-fold higher for polyadenine (polyA$_N$) than for mixed-sequence oligonucleotides ($N_{min} = 6.6$ vs 3.7, respectively) (Supplementary Fig. S13). Therefore, short, mixed-sequence oligonucleotides exhibit a higher propensity towards coacervation than less prebiotic, homopolymeric strands. Conversely, increasing the oligonucleotide charge, by means of phosphate groups on the 5′ and 3′ ends, causes the formation of clusters of coacervates and solid-like aggregates (Supplementary Fig. S14), potentially due to the additional electrostatic interactions with the more exposed, terminal phosphate groups.

Overall, our results indicate that, even in a prebiotic scenario where heterogeneous RNA and DNA oligomers were present together with short peptides, phase separation likely occurred and potentially impacted the chemistry taking place at the dawn of a nucleic acid-peptide world.

### Peptides interact more with RNA than DNA

To elucidate the distinct features of the interactions between peptides and DNA or RNA strands and rationalise the unexpected different salt and thermal stabilities of the resulting coacervates, we carried out atomistic force-field simulations of four mixtures: $R_3$/DNA$_8$, $R_4$/DNA$_8$, $R_3$/RNA$_8$ and $R_4$/RNA$_8$. Our models contain eight single-stranded (ss) oligonucleotides with thirty-six Arg peptides in explicit solvent and ions. For each mixture, we analysed the trajectories to quantify the frequency of intermolecular contacts between Arg peptides and oligonucleotides.

Arginine is known to interact with RNA through multiple modes[42], yet any distinction between RNA and DNA oligomers to undergo coacervation has never been explored, due to the focus on probing the differences between double-stranded and single-stranded DNA (for their significance in genomic function)[43,44], and the outdated assumption that RNA preceded DNA on early Earth[45].

We identified three main intermolecular interaction modes: ionic (here defined as non-hydrogen bonding contacts between the positively charged sidechain of Arg and the backbone phosphate group in both oligonucleotides), hydrogen bonding, and stacking, including π-π stacking and cation-π interactions between the positively charged Arg sidechain and the nucleobases in both oligonucleotides (Fig. 2a). Across all interaction classes, RNA$_8$ consistently forms more contacts with Arg peptides than DNA$_8$, with differences being most pronounced in stacking interactions (96% increase in contact points for RNA$_8$ over DNA$_8$) and hydrogen bonding via the nucleobase (Fig. 2b, Supplementary Figs. S15 and S16, and Supplementary Tables S7 and S8). Despite their relatively low frequency, the enhanced strength of cation-π interactions, demonstrated through quantum mechanical calculations of model systems relative to ionic or hydrogen bonding in aqueous media[46], suggests that even minor variations in their occurrence can have a significant energetic impact. As such, the higher number of stacking interactions present in the RNA$_8$ systems are expected to be a key contributor to the higher thermodynamic stability of RNA$_8$-based coacervates, as also shown by the studies on the thermal stability of coacervates (Fig. 1c).

The higher frequency of intermolecular interactions observed for RNA$_8$ over DNA$_8$ are attributed to conformational differences between the two nucleic acids, likely due to the additional hydroxyl group in the sugar moiety of ribonucleotides. In contrast to DNA$_8$, RNA$_8$ adopts a more expanded, unfolded structure with Arg peptides (Fig. 2c), which enables its nucleobases to engage more readily in intermolecular hydrogen bonding and stacking interactions with Arg residues. Our all-atom simulations reveal that the higher propensity of RNA$_8$ versus DNA$_8$ to acquire an expanded, unfolded structure within the coacervate phase results in an increased density of intermolecular interactions and an enthalpic gain for coacervation[47]. This observation aligns with previous structural analyses that reveal stronger and more frequent π−π contacts of Arg with RNA nucleobases than with DNA

nucleobases[48]. Several other factors likely contribute, such as uracil's weaker stacking interactions with other nucleobases compared to thymine[49] and the higher tendency of DNA to adopt compact helical conformations[50].

The total number of intermolecular contacts that RNA$_8$ or DNA$_8$ form with Arg peptides (Fig. 2d and Supplementary Fig. S17, and Supplementary Table S9) correlates well with experimentally observed phase separation propensity (Fig. 1). Indeed, simulations on the $R_3$/DNA$_8$ system (the only mixture that does not form coacervates) show the lowest number of intermolecular contacts. Notably, elongating the peptide chain by one Arg residue (from $R_3$ to $R_4$) results in an 18 and 16% increase in hydrogen bonding and ionic interactions, respectively (Supplementary Fig. S15). This increase in the total number of interactions for the $R_4$/DNA$_8$ system aligns well with our experimental finding that $R_4$ is the minimum peptide length required for coacervation with DNA$_8$ (Fig. 1d). In contrast, the $R_3$/RNA$_8$ mixture has a similar number of intermolecular contacts to $R_4$/DNA$_8$ due to the more abundant hydrogen bonding and stacking interactions characteristic of RNA$_8$, and thus undergo coacervation. This observation and the fact that excess peptide remains in the simulation box at equilibrium (Fig. 2e and Supplementary Fig. S18, and Supplementary Table S10) suggests that the intrinsic physicochemical differences between DNA and RNA can explain their observed different sensitivity to peptide length.

Our simulations reveal striking differences in how Arg peptides interact with DNA and RNA, which allow us to explain the macroscopic differences in the phase separation behaviour that we observed experimentally. RNA$_8$ exhibits a notably higher frequency of stacking and hydrogen bonding with Arg peptides than DNA$_8$ (Fig. 2b), which likely underpins the increased resilience to both salt concentration (Fig. 1a) and temperature (Fig. 1c), and thus the thermodynamic stability of RNA-based coacervates.

### Nucleic acids quickly diffuse in primitive DNA coacervates

Model coacervates can increase the local concentration of dilute solutes, including oligonucleotides, and potentially facilitate replication reactions[18,51]. Whether the differences observed in DNA- and RNA-based coacervates would influence their ability to recruit peptides and oligonucleotides is unknown. The partition coefficients for several fluorescently labelled probes (FITC-R$_8$, FAM-DNA$_8$ and FAM-RNA$_8$) was measured by confocal microscopy (Supplementary Table S11). FITC-R$_8$ exhibited a 1.5 times higher partition coefficient in $R_8$/RNA$_8$ than in $R_8$/DNA$_8$ coacervates (Fig. 3a), likely due to the greater number of contacts between Arg peptides and RNA (Fig. 2b). FAM-DNA$_8$ and FAM-RNA$_8$ partitioned similarly in $R_4$/DNA$_8$ coacervates, yet $R_4$/RNA$_8$ coacervates recruited 1.3-fold more FAM-RNA$_8$ instead of FAM-DNA$_8$ (Fig. 3b). The difference in partition coefficients for RNA-based coacervates likely results from the higher energetic cost of recruiting a conformationally rigid and less interacting DNA probe into RNA coacervates[52]. Importantly, oligonucleotides that are too short to undergo phase separation are efficiently recruited in primitive coacervates (Supplementary Fig. S19).

Diffusivity within coacervates is also key to support nucleic acid reactivity, e.g., ribozymatic functionality and non-enzymatic RNA polymerisation. Specifically, coacervates with long RNA strands (>50 nucleotides)[17] or Arg homopeptides[18] are known to inhibit ribozymatic activity due to their higher viscosity. We thus characterised the fluidity of primitive coacervates by using fluorescently-labelled oligonucleotides and Fluorescence Recovery After Photobleaching (FRAP) measurements. The recovery time measured in FRAP of coacervate droplets is affected by at least three parameters: viscosity of the coacervate phase, size (length) of the FRAP probe and attractive interactions between the probe and the coacervate scaffold.

Firstly, we explored the influence of probe size on its diffusion within minimal coacervates, by employing three model Cy3-labelled

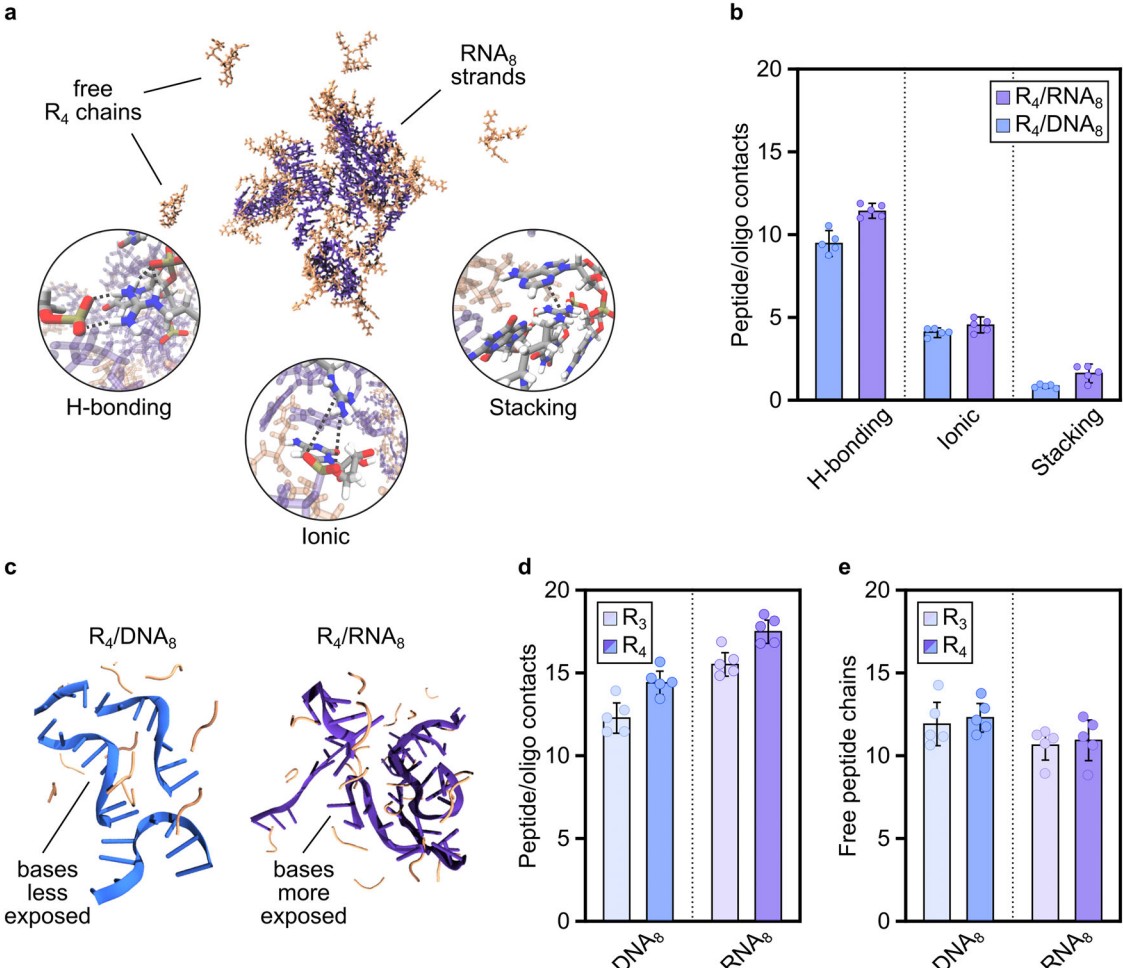

**Fig. 2 | Computational investigations reveal contact modes and frequency of interactions in peptide/nucleic acid coacervates. a** Representative atomistic force-field simulation snapshot of the $R_4$/$RNA_8$ mixture, showing a cluster of RNA and peptide, and unbound peptide in excess. Inset shows the interaction modes, e.g., hydrogen bonding, ionic interactions, and cation-π/π-π stacking. The simulations were performed with OpenMM 8.1.2, leveraging the CUDA platform in mixed precision mode, using the Amber14SB force field for peptides, OL3 parameters for RNA, and bsc1 for DNA. See Supplementary Information for more details. **b** Comparison between the number of DNA and RNA interactions with arginine peptides (per frame, per nucleotide), separated into three categories: hydrogen bonding, ionic interactions and stacking. **c** Simplified rendering of $R_4$/$DNA_8$ (blue) and $R_4$/$RNA_8$ (purple) clusters, showing the helical, structured conformation acquired by DNA strands and the more disordered folding acquired by RNA strands, which leaves ribonucleotides exposed to interact with peptides. **d** Number of peptide/oligonucleotide contacts (all modes of interaction), which represents the total number of intermolecular contacts that one molecule of $RNA_8$ or $DNA_8$ forms with $R_3$ or $R_4$. **e** Excess peptide remaining in the simulation box at equilibrium for $RNA_8$ or $DNA_8$ coacervates with $R_3$ or $R_4$. R = Arginine, $DNA_8$ = 8-deoxyribonucleotide-long mixed sequence ($(ACTG)_2$), $RNA_8$ = 8-ribonucleotide-long mixed-sequence oligonucleotide ($(ACUG)_2$). $n = 5$ for data in (**b**), (**d**) and (**e**); mean and SD values are provided. All plotted values can be found in Supplementary Tables S7–S10. Source data are provided as a Source Data file.

probes of different lengths (Fig. 3c). Empirical recovery times (τ) for all probes are 5–95 s, which indicates extraordinarily high mobility within primitive coacervates (Supplementary Table S12) compared to previously studied coacervate models which display recovery times on the scale of several minutes[17]. In most cases, we observed an incomplete recovery after photobleaching, likely due to the high partitioning of the probe (Supplementary Table S11) and its slower exchange postbleaching between dilute and coacervate phases, beside dye photofading[53,54].

As expected, the empirical recovery time is proportional to the length of the probe and that of the coacervate components, which is correlated to the viscosity of the coacervate (Fig. 3d, Supplementary Figs. S20 and S21). Interestingly, peptide length has a stronger effect on probe diffusion within coacervates than oligonucleotide length: the recovery time of a 8-nucleotide long probe in $R_4$/$DNA_8$ coacervates is nearly half of that observed in $R_4$/$DNA_{16}$ coacervates (10 s and 18 s, respectively), whereas the use of $R_8$ instead of $R_4$ for coacervation

induces a 3-fold increase in the recovery time of the probe. These results suggest that coacervate stability is more strongly dependent on peptide, rather than oligonucleotide length (Supplementary Table S3).

Secondly, we investigated the potential effect of coacervate composition on probe diffusion. The empirical recovery times in $R_4$/$DNA_8$ and $R_4$/$RNA_8$ coacervates are consistent with low viscosity liquid phases. Yet, we found that the probe is strikingly more mobile in DNA-based than in RNA-based coacervates (empirical recovery times of 10 s and 62 s, respectively) (Fig. 3e and Supplementary Fig. S22). As the probe sequence was chosen to minimise base-pairing with the nucleic acid scaffold, the significant difference in recovery times cannot be attributed to interactions between the probe and the matrix alone. Instead, the stronger and more frequent interactions between RNA and Arg peptides (Fig. 2) result in coacervates with higher viscosity, resulting in a slower probe diffusion than their DNA-based analogues. Similarly, coacervates made up of adenosine-rich oligonucleotides that interact less with Arg peptides (Supplementary Table S8) result in

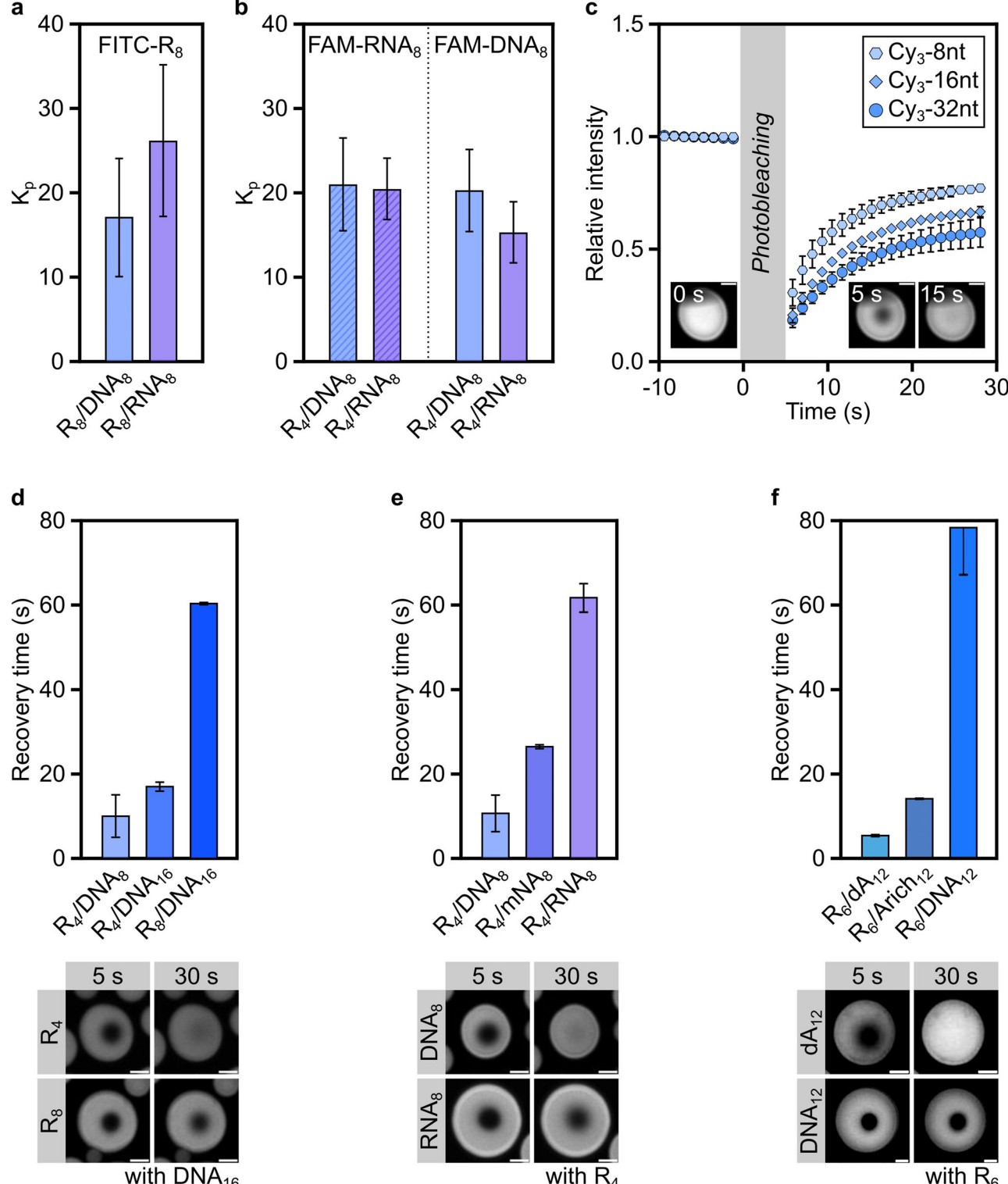

**Fig. 3 | Composition of primitive peptide/NA coacervates modulates their biophysical properties.** **a** Partitioning of a labelled peptide (1% FITC-R$_8$) in peptide/nucleic acid coacervates. $n = 3$. **b** Partitioning of labelled oligonucleotides (1% FAM-DNA$_8$ or 1% FAM-RNA$_8$) in peptide/nucleic acid coacervates. $n = 3$. **c** Example of FRAP profiles for the investigated peptide/nucleic acid coacervates. The fit of three probes in R$_4$/DNA$_8$ coacervates is included for clarity. **d** Recovery time for probe Cy3-8nt in coacervates of varying peptide and DNA length: R$_4$/DNA$_8$, R$_4$/DNA$_{16}$ and R$_8$/DNA$_{16}$. A 20 mM:5 mM [Arg]:[nucleotide] ratio was used for these experiments. **e** Recovery time of probe Cy3-8nt in coacervates comprising R$_4$ and DNA$_8$, RNA$_8$ or a DNA$_8$:RNA$_8$ (1:1 ratio) mixture (mNA$_8$). A 20 mM:5 mM [Arg]:[nucleotide] ratio was used for these experiments. **f** Recovery time of probe Cy3-8nt in coacervates comprising R$_6$ and DNA$_{12}$, dA$_{12}$ or an A-rich sequence, Arich$_{12}$. A 20 mM:10 mM [Arg]:[nucleotide] ratio was used for these experiments. Scale bar: 2 μm. R = Arginine, nt = nucleotide, DNA$_8$ = 8-deoxyribonucleotide-long mixed-sequence oligonucleotide ((ACTG)$_2$), DNA$_{16}$ = 16-deoxyribonucleotide-long mixed-sequence oligonucleotide ((ACTG)$_4$), RNA$_8$ = 8-ribonucleotide-long mixed-sequence oligonucleotide ((ACUG)$_2$), τ = empirical recovery time. $n = 3$ for data (**a**) and (**b**); $n = 3$ droplets for data in (**c**)–(**f**); mean and SD values are provided. K$_p$ values plotted in (**a**) and (**b**) can be found in Supplementary Table S11; details of the exponential fit used to extract τ in (**d**)–(**f**) are in Supplementary Table S12. Source data are provided as a Source Data file.

faster recovery than those made of mixed-sequence oligonucleotides (Fig. 3f). Notably, the addition of salt or short non-coacervating DNA oligonucleotides to primitive coacervates results in lower empirical recovery times, and thus lower viscosity, by weakening the interactions between the coacervate components (Supplementary Fig. S23 and Supplementary Table S12).

Overall, our findings showcase the remarkable low viscosity of primitive coacervates compared to artificial systems proposed so far as models for primitive cells. The lower tendency of DNA to interact with peptides compared to RNA could have led to the emergence of coacervates with extraordinary fluidity and fast diffusion of partitioned peptides and nucleic acids, a seeming requirement[17] for prebiotic RNA activity. Notably, nucleic acid-based coacervates comprising both RNA and DNA would have exhibited remarkable fluidity, thanks to the ability of DNA to mitigate RNA-peptide interactions without impacting coacervate stability (Fig. 1d), hinting at an early synergy between RNA and DNA.

### Primitive coacervates enable RNA polymerisation

The RNA-peptide world hypothesis posits an evolutionary period in which primitive lifeforms relied heavily on the catalytic properties and information carrying capabilities of RNA alongside peptides[55,56]. Non-enzymatic RNA replication is thought, however, to have played a pivotal role prior to the rise of an RNA replicase (or ribozymes with an analogous function) to effectively replicate the RNA genome[57]. We thus investigated whether primitive coacervates could have supported RNA folding and function, e.g., non-enzymatic RNA polymerisation.

As a preliminary assessment, we used a well-studied split version of the Broccoli aptamer[36,58] in solutions that contained $R_4/RNA_8$, $R_4/DNA_8$ or $R_4/DNA_{16}$ coacervates, and measured the fluorescence intensity of the bound fluorogenic probe, 3,5-difluoro-4-hydroxybenzylidene imidazolinone (DFHBI) (Supplementary Figs. S24 and S25). The fluorescence of the light-up aptamer, and thereby its secondary structure, was fully preserved within $R_4/DNA_8$ coacervates, but only partially maintained in $R_4/RNA_8$ coacervates or in coacervates comprising sufficiently long DNAs such that non-homogeneous partitioning was observed (Fig. 4a, b). Interestingly, the fluorescence in $R_4/DNA_{16}$ was not only attenuated, but also concentrated on the edge of the droplets, suggesting a slower diffusion of the aptamer within the coacervates. These findings indicate that coacervates made of short DNA oligonucleotides and Arg peptides – characterised by weaker and less abundant interactions, and hence remarkably enhanced mobility, than their RNA-based counterparts (Fig. 2b) – preserve nucleic acid folding more efficiently.

Non-enzymatic genome copying is thought to be a crucial process in the emergence and evolution of early lifeforms, particularly in the rise of functional RNA sequences[57]. Although it was shown that coacervates comprising synthetic polycations, including poly-allyldiammonium chloride, supported template-directed RNA elongation, highly viscous coacervates comprising polyarginines and RNA oligonucleotides ($R_{10}/rA_{11}$) inhibited RNA reactivity[18]. Yet, encouraged by the extraordinary fluid-like behaviour of DNA-based coacervates, we investigated whether the efficiency of non-enzymatic RNA polymerisation would be preserved in the presence of primitive coacervates (Fig. 4c).

Primer extension reactions are useful model experiments that reflect the first step of non-enzymatic genome replication. Here, a fluorescently labelled primer hybridised to a complementary template (containing a GG overhang within the template strand) (Supplementary Table S2) was added to a solution of DNA-based or RNA-based coacervates. The primer was designed with the same length and composition as the probes used in partitioning and FRAP experiments (Fig. 3) to achieve similar accumulation and mobility within peptide/oligonucleotide coacervates. Primer extension reactions were initiated upon addition of the activated dinucleotide[59] and $MgCl_2$ and assessed

by gel electrophoresis (Fig. 4d and Supplementary Fig. S26 and Supplementary Tables S13 and S14). Control experiments performed in the presence of the *host* oligonucleotide (i.e., involved in coacervation, but not engaged in primer extension), but without peptide, showed that the reaction is as efficient in the presence of bystander oligonucleotides as in their absence (Supplementary Fig. S27), as long as no complementarity exists between host and primer/template oligonucleotides.

Due to the reactivity of amines towards the activated dimer (Supplementary Fig. S28)[60,61], we expected Arg peptides to compromise primer extension. We thus evaluated the efficiency or primer extension in the presence of $R_6$, adding NaCl to prevent phase separation. Regardless of the concentration of $R_6$, primer extension was 1.5 times less efficient in the presence of peptide than without after 24 h (Supplementary Table S14 and Supplementary Figs. S29 and S30). These findings suggest that non-enzymatic template-directed RNA elongation may have been inefficient in crowded prebiotic settings in which reactive peptides, prone to degrade the activated dimer, were abundant. Consequently, we wondered whether, upon coacervation, peptides would have a less detrimental effect on non-enzymatic RNA polymerisation.

When primer extension was performed in the presence of $R_6/dA_{12}$ coacervates, the reaction efficiency was mostly restored (≈90% extended primer after 24 h) (Figs. 4d, 4e, Supplementary Figs. S29 and S30). Similar yields of primer extension were obtained for a different primer-template system (Supplementary Fig. S31) to support the generalisability of our findings. As the reaction can take place both in the coacervate and the dilute phases, all reported results refer to the overall reaction yield in the presence or absence of coacervates. Still, as previously reported[62], coacervation likely enhances the rate of primer extension in the dilute phase by sequestering peptides that would otherwise inhibit the reaction. As expected, by changing the charge ratio between *coacervating* peptide and oligonucleotide, thus increasing the excess of free peptide in solution (without affecting the composition of the dense phase, Supplementary Fig. S32), the reaction efficiency partially decreased (63% after 24 h) (Supplementary Fig. S33).

We next sought to understand the relationship between the viscosity of primitive coacervates and the functionality of their guest RNA strands (Fig. 4f). We thus explored the suitability of coacervates comprising host oligonucleotides of different composition, length and sequence, for non-enzymatic primer extension. When the reaction was performed in relatively viscous coacervates composed of longer polyadenine strands ($dA_{16}$ in lieu of $dA_{12}$), primer extension was still observed (76% after 24 h), albeit with lower efficiency (Fig. 4e and Supplementary Fig. S34). Similarly, when RNA ($rA_{12}$) was used for coacervation, 64% extended primer was detected (Fig. 4e and S34). Coacervates comprising mixed DNA and RNA oligonucleotides ($mA_{12}$, i.e., $dA_{12}:rA_{12}$ 1:1 ratio) showed an intermediate efficiency of primer extension (73% after 24 h) (Fig. 4e and Supplementary Fig. S35). These results confirm that the viscosity of coacervates, which is a result of their composition (Fig. 3d–f), directly effects RNA chemistry (Fig. 4f).

Based on the propensity of each nucleotide to interact with Arg peptides (Supplementary Table S8), we hypothesised that mixed-sequence oligonucleotide coacervates would exhibit diminished capability of supporting RNA polymerisation. As expected, we observed a high level of inhibition in $R_4/DNA_{12}$ coacervates (36% after 24 h) (Supplementary Fig. S36). Increasing the adenine content in the host oligonucleotide sequence ($Arich_{12}$), which lowered the viscosity of the resulting coacervates, resulted in higher primer extension yields (41% after 24 h) (Fig. 4e and Supplementary Fig. S37). Elongating the peptide ($R_6$ in place of $R_4$) also increased the yield of primer extension (47% after 24 h) (Supplementary Fig. S36) despite the enhanced viscosity of the resulting coacervates (Supplementary Table S12), which suggests that the stability of primitive coacervates also plays a role in enabling

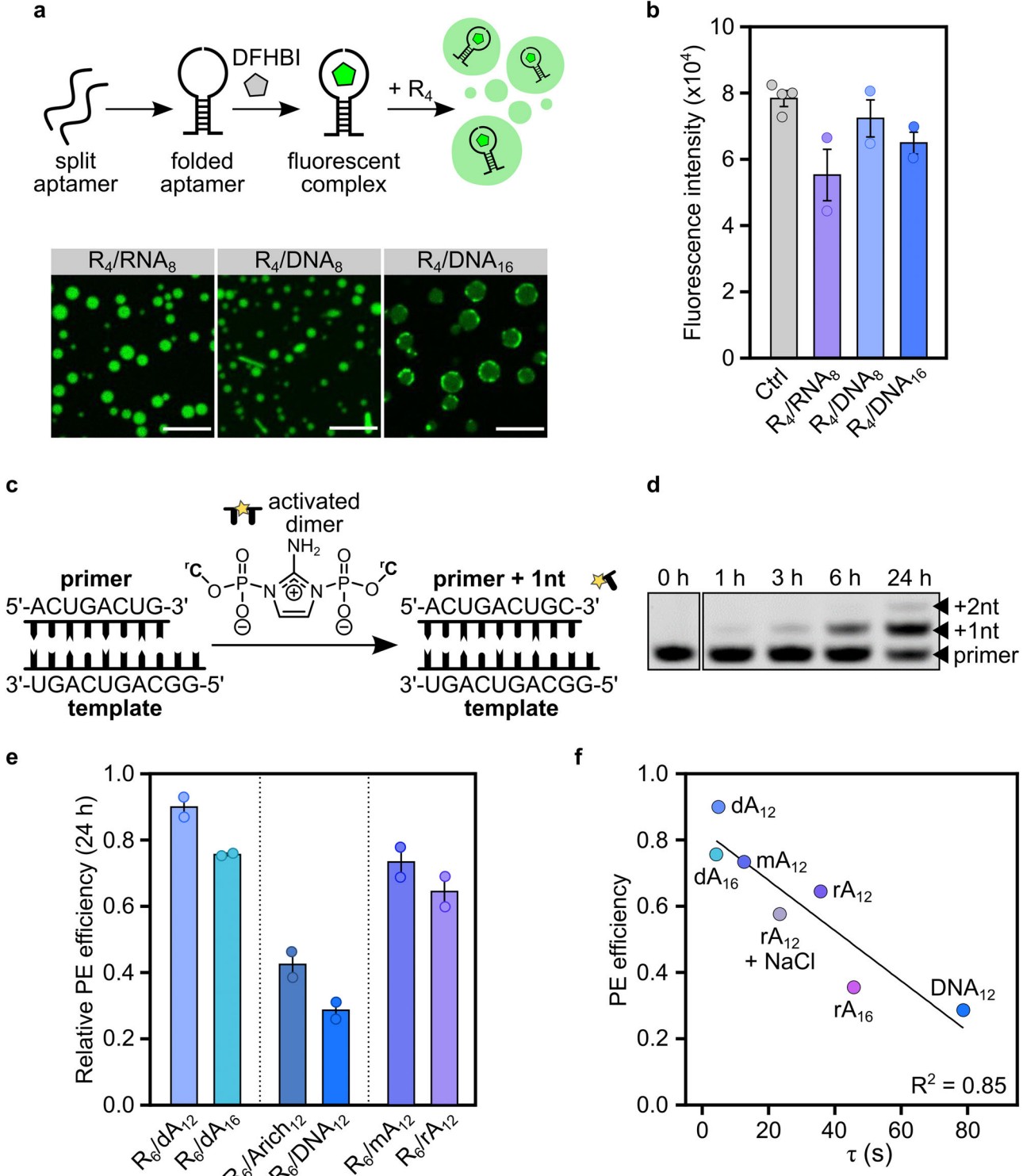

**Fig. 4 | Primitive coacervates are compatible with prebiotic RNA chemistry and functionality. a** Schematic representation and confocal micrographs of the split Broccoli aptamer reconstitution in primitive coacervates. Scale bar: 10 μm. **b** Total DFHBI emission in Broccoli aptamer samples after addition of $R_4$ (to trigger coacervation). In the 1-phase control, no oligonucleotide strand other than the split aptamer was present, and no peptide was added. **c** Schematic representation of the PE reaction. For all reactions involved in this study, 5 mM $Mg^{2+}$ was used as catalyst. **d** Representative denaturing polyacrylamide gel image of PE in $R_6$/$dA_{12}$ coacervates at different time points. **e** Relative PE efficiency as a function of oligonucleotide composition or length; for a primer/template system showing no complementarity with the host strand. Error bars represent S.E.M from at least two independent experiments. **f** Relationship between PE efficiency and empirical recovery time in $R_6$-based coacervates (20 mM:10 mM [Arg]:[nt]); for a primer/template system showing no complementarity with the host strand. R = Arginine, nt = nucleotide, PE = primer extension, $dA_{12}$ = 12-deoxyribonucleotide-long polyadenine oligonucleotide, $dA_{16}$ = 16-deoxyribonucleotide-long polyadenine oligonucleotide, $rA_{12}$ = 12-ribonucleotide-long polyadenine oligonucleotide, $mA_{12}$ = $dA_{12}$:$rA_{12}$ 1:1 ratio, $Arich_{12}$ = 12-deoxyribonucleotide-long A-rich-sequence oligonucleotide (AAGTAAAG-TAAA), $DNA_{12}$ = 12-deoxyribonucleotide-long mixed-sequence oligonucleotide ((ACTG)$_3$), τ = empirical recovery time. *n* = 2 for data in (**b**), (**e**) and (**f**); mean and SD values are provided. Normalised and un-normalised yields for primer extension at different time points can be found in Supplementary Table S14, including results for $R_4$-based coacervates. Source data are provided as a Source Data file.

efficient RNA polymerisation. Since coacervates with higher CSCs have larger volume fractions at a given salt concentration[63], we reason that the coacervate phase, despite its minimal volume fraction (<1%), significantly influences the efficiency of RNA polymerisation. Interestingly, although RNA polymerisation is fully suppressed in coacervates with a host RNA sequence that is fully complementary to the guest RNA template, host DNA coacervates tolerate complementary guest RNA and a degree of primer extension (29% with complementary RNA template versus 47% for non-complementary RNA template) (Supplementary Fig. S38). This finding may be due to the higher stability of RNA:RNA duplexes relative to DNA:RNA hybrids[64,65].

Overall, although ribozyme activity and template-directed RNA polymerisation are inhibited in model polyarginine/RNA coacervates[17,18], we show that primitive DNA-based coacervates can fully preserve RNA folding and efficiently support non-enzymatic RNA primer extension, namely due to their highly fluid nature. Notably, the remarkable differences in stability, fluidity and functionality between DNA and RNA coacervates are suggestive of a nucleic acid-peptide world scenario in which all precursors of the central dogma of biology could have played distinct, key biochemical roles with potential to support the emergence of life.

## Discussion

Complex coacervates, formed upon liquid-liquid phase separation of oppositely charged polymers, have long been suggested as models of primitive cells. Yet, the low prebiotic plausibility of the coacervating components studied, commonly designed to maximise coacervate stability or functionality, led to the notion that the emergence of coacervates succeeded the synthesis of long, sequence-specific, functional polymers (i.e., homopeptides and ribozymes). Our work challenges this assumption by demonstrating that heterogeneous oligonucleotides spontaneously undergo phase separation with peptides to generate primitive coacervates, which likely impacted prebiotic RNA chemistry.

The prebiotic plausibility of short peptides, RNA and DNA oligomers, and their intertwined role in the central dogma of biology, suggest their cooperation, likely due to co-localisation, early on the evolutionary timeline. Our work shows that primitive coacervates can be generated by liquid-liquid phase separation of short oligonucleotides and peptides (i.e., peptide dimers and trimers, RNA and DNA octamers). These findings suggest that compartmentalisation via coacervation could have occurred simultaneously to the early stages of non-coded amino acid and nucleotide polymerisation. In contrast with peptide-peptide coacervates, these minimal nucleic acid-peptide coacervates showcase enhanced stability to high concentration mismatch of their components and elevated salinity, thus loosening the chemical constraints on the prebiotic environments that could have accommodated coacervation.

The seemingly inevitable tendency of short heterogeneous oligonucleotides and peptides to undergo phase separation suggests that coacervates were unlikely selected as a fitness advantage at a late evolutionary stage, but rather were a consequence of prebiotic molecular composition in an early nucleic acid-peptide world.

Primitive coacervates can be effectively described by all-atom simulations of peptide/nucleic acid condensation. Mixtures comprising RNA oligonucleotides are characterised by a higher number of contacts between arginine residues and nucleotides compared to DNA-based counterparts, likely due to the more extended and less structured conformation acquired by RNA over DNA upon phase separation. Although both nucleic acid/peptide coacervates exhibit enhanced stability and fluidity over previously reported models, we show that the chemical diversity of RNA and DNA is mirrored in the diverse properties (stability, fluidity and functionality) of the resulting coacervates.

Our work offers a set of guiding molecular principles to generate models of biomolecular condensates. Fine-tuning the fluidity of coacervates by modulating the DNA-to-RNA ratio enabled us to explore RNA chemistries (e.g., ribozymatic activity) that were, until now, considered incompatible with primitive coacervates. More broadly, this study highlights how small molecular differences in oligonucleotide or peptide composition or length can have remarkable macromolecular effects on the material properties of the resulting coacervates. The possibilities to explore the effect of other molecular alterations, including employing non-canonical nucleotides or amino acids, or building blocks with opposite chirality, on coacervate functionality are arguably unlimited.

Non-enzymatic RNA copying would have been important prior to the rise of a ribozyme capable of replication. Testing its compatibility with primitive coacervates was thus critical. We found that coacervate stability, charge and, most importantly, fluidity are key factors that control the chemical copying of RNA, with a high degree of predictability. The unexpected observation that DNA-based coacervates more efficiently preserve RNA secondary structures and support non-enzymatic RNA polymerisation suggests that, before the genetic takeover of RNA, DNA oligonucleotides might have played a key role in compartmentalisation by enabling the emergence of coacervates compatible with primitive RNA activity. All in all, the seemingly inevitability of coacervation invites us to revisit prebiotic chemistry for its compatibility and efficacy in phase-separated environments; and the unique ability of primitive DNA-containing coacervates to efficiently preserve RNA folding and support RNA functionality offers a plausible trajectory for the early evolution of primitive cells with sequence-dependent phenotypes.

## Methods

### Materials

Reagents were purchased from Merck and Thermo Fisher and used without further purification unless otherwise stated. Polyuridylic acid (polyU) potassium salt (MW 600–1000 kDa, ~2000–3200 bases) was purchased from Merck. N-benzoyl-dA, N-isobutyryl-dG, N-acetyl-dC and dT phosphoramidites, and 2′-O-TBDMS protected, N-benzoyl-rA, N-isobutyryl-rG, N-acetyl-rC and rU phosphoramidites, and 6-FAM amidite (CLP-9777) were purchased from ChemGenes (Wilmington, MA). Oligonucleotides were purchased from Integrated DNA Technologies (IDT) and Eurofins or synthesised in-house when indicated. Peptides were purchased as TFA salts from GenScript or synthesised in-house when indicated. Sep-Pak C18 classic cartridge was purchased from Waters (Milford, MA). Water coming into contact with DNA/RNA oligomers was 18 MΩ grade.

Fmoc solid phase peptide synthesis (Fmoc-SPPS) was carried out on an induction heating-assisted PurePrep® Chorus synthesiser (Gyros Protein Technologies) pressurised with 4.5 $N_2$ and equipped with two independent reaction vessel slots with both induction heating and a UV-monitoring detector. Reverse-phase high-performance liquid chromatography (RP-HPLC) purifications on peptides were performed using an Agilent semi-preparative HPLC system equipped with a 1260 Infinity II binary pump, 1260 Infinity II variable wavelength detector with 3 mm preparative cell, and a 1290 Infinity II preparative open-bed sampler/collector with a 20 mL injection loop on a ReproSil Pur 120 C18-AQ 250 ×25 mm 5 μm particle size column from DrMaisch GmbH. Purification of oligonucleotides was performed using a DNApac™ PA200 column with a Vanquish™ analytical purification high-performance liquid chromatography (HPLC) system. DNA and RNA melting temperatures and base pairing probabilities were assessed using NUPACK 4.0 (https://www.nupack.org/). pH monitoring was performed using a Mettler Toledo FiveEasy pH meter and adjustments were made with aqueous solutions of NaOH or HCl as appropriate. The turbidity of mixtures was determined using a BMG Labtech

CLARIOstar[plus]. Concentrations were calculated using the Beer-Lambert equation (molar extinction coefficients were estimated using the OligoAnalyzer™ Tool (IDT)).

Coacervates were imaged using a Nikon Eclipse TS2 inverted epifluorescence microscope equipped with a Moment A21K635003 camera (0.63× adaptor) and a 60× oil immersion objective. Alternatively, coacervates were imaged using a Zeiss Axio Observer Z1 spinning disk confocal microscope equipped with a Yokogawa CSU confocal head and a 63× oil immersion objective. Images were processed using Fiji (http://rsb.info.nih.gov/ij/). Thermal studies were performed using a home-built Nikon Eclipse Ti-E inverted microscope equipped with a 20× objective lens (Nikon, Plan Fluor, N.A. 0.75) and a Grasshopper3 GS3-U3-23S6M camera (Point Gray Research). The illumination was provided by single-colour light-emitting diodes (LEDs) using a filter set for Texas Red. Temperature ramps were performed using a custom-built script, enabling precise manipulation of the instrument in terms of time, temperature, and illumination as required. FRAP experiments were performed using a Leica TCS SP5 laser scanning confocal microscope (Cavendish Laboratory, Cambridge) equipped with an HCX PL Apo 40× DRY (NA 0.85) objective lens and a HeNe laser (633 nm, 10 mW). DNA and RNA oligonucleotides (ONs) were synthesised using an ABI-394 DNA synthesiser. UV measurements on oligonucleotides were taken at 260 nm using an Agilent BioTek Epoch Microplate Spectrophotometer, reading each sample at least 3 times and correcting each value by a blank measurement. Polyacrylamide gels were imaged on an Amersham TYPHOON using the Cy2 laser at 25-50 μm pixel size.

Average and standard deviation values refer to $n \geq 3$ replicates. Statistical significance was determined using unpaired t-tests (ns $P > 0.05$; $*P \leq 0.1$; $**P \leq 0.01$; $***P \leq 0.001$; $****P \leq 0.0001$).

**Stock solutions.** Peptide stocks were prepared in MilliQ water at a concentration of 100 mM based on the molecular weight of the TFA salt. The pH of $E_{10}$ (glutamic acid decamer) was adjusted with ammonia for complete dissolution. Solutions were sonicated, stored at −20 °C, and vortexed for 1 min before use. Single-strand DNA and RNA oligonucleotide stocks were prepared in DNase/RNase-free water at a strand concentration of ~1 mM. To facilitate the solubilisation of the oligonucleotides, solutions were heated to 50 °C for 5 min and cooled down to room temperature before the measurement. The concentration was checked on a diluted solution (250–500×), measuring the absorbance at 260 nm.

**Coacervate preparation.** Coacervates were prepared in a 10–100 μL scale by adding, respectively, MilliQ water, HEPES buffer (from a 500 mM stock buffer solution, pH 7.4), DNA or RNA (~1 mM oligonucleotide stock) and peptide (100 mM stock). Aptamer and fluorescent probes were added at last unless otherwise stated. Mixing was done by gently tapping the microtube to avoid reducing droplet size for imaging. Mixtures were assessed by light microscopy to confirm the presence of liquid droplets. Note: in FRAP and aptamer experiments, the peptide was added at last to enable the fast incorporation of fluorescent dyes.

**Preparation of observation chambers.** A passivated glass coverslip #1.5 was used as the observation surface in all experiments. Glass passivation was performed to prevent wetting. A 5 wt% solution of partially hydrolysed polyvinyl alcohol (PVA, 13-23k) was spread on top of clean coverslips and let adsorb for 1 h inside a covered petri dish. The coverslips were rinsed thoroughly with distilled water and once with MilliQ before being dried with compressed air. For long imaging experiments (FRAP, $K_p$ measurements, thermal ramps), 2–6 μL chambers were prepared using double-sided 3 M tape (GPT-020F, 0.2 mm) and a hole-punch (2–4 mm Ø) and sealed using 10 mm Ø coverslips to prevent evaporation.

**Fmoc solid-phase peptide synthesis (Fmoc-SPPS).** $R_4$ and $R_2$ were synthesised according to previously published SPPS procedures by using Fmoc-protected amino acids[66]. Specifically, 0.50 mmol of commercially available, pre-loaded Wang resin was added to a plastic reactor equipped with a fritted plastic insert. The resin was allowed to swell in DMF for 30 min. For the deprotection step, 20% piperidine in DMF (5 mL/0.5 mmol) was added to the resin. The resin was left to react for 2 min before the removal of the solvent. The treatment was repeated with 20% piperidine in DMF and left shaking for 15 min. The solvent was removed, and the resin was washed with DCM/DMF (5 × 5 mL). For the coupling step, Fmoc-protected amino acids (Fmoc-AA-OH) (3.0 equiv. relative to the resin loading) were dissolved in dry DMF. A solution of 2-(1H-benzotriazol-1-yl)-1,1,3,3-tetramethyluronium hexafluorophosphate (HBTU, 3.8 equiv.) was added to the Fmoc-AA-OH solution, followed by N,N-Diisopropylethylamine (DIPEA, 6.0 equiv.), and added to the resin. The resulting mixture was agitated on a laboratory shaker for 45 min. DMF washes of the resin (5 × 5.0 mL) were performed before deprotection. Cycles of coupling and deprotection steps were performed to obtain the desired peptide sequence. After the final Fmoc removal, the resin was washed with DMF (3×), DCM (3x), and MeOH (3×) and left to dry under a high vacuum overnight.

For cleavage, the resin was treated with the cleavage solution (trifluoroacetic acid (TFA):$H_2O$:triisopropyl silane 95:2.5:2.5 volume ratio) for 2 h. TFA-peptide solutions were collected, and the resin was washed with TFA (2 × 3 mL). The collected fractions were concentrated under nitrogen flow and added to cold diethyl ether, leading to the precipitation of the peptide. The precipitate was centrifuged for 5 min at $5000 \times g$ and washed with cold diethyl ether (10 mL). The resulting peptide was dissolved in acetonitrile:water 1:5 (10 mL) and lyophilised. Peptides were purified by RP-HPLC. Elution was performed at a flow rate of 10 mL/min using a linear gradient of acetonitrile and ultrapure water (both containing 0.1% LCMS grade formic acid). The gradient ranged from 20% to 80% acetonitrile over 1 h. UV absorption at 220 nm and 254 nm was used to monitor the collection of unprotected peptides. Fractions containing the target product were identified by mass spectrometry and lyophilised.

$R_4$: [1]H-NMR (500 MHz, $D_2O$) δ (ppm) 3.95-3.86 (m, 3H), 3.19-3.10 (m, 8H), 1.95–1.79 (m, 8H), 1.73–1.51 (m, 8H).

$R_2$: [1]H-NMR (500 MHz, $D_2O$) δ (ppm) 4.44-4.35 (m, 1H), 4.09 (t, J = 6.4 Hz, 1H), 3.25 (t, J = 6.8 Hz, 2H), 2.58–2.44 (m, 2H), 2.28–1.61 (m, 6H).

**Synthesis of imidazolium-bridged dicytidyl dimer intermediate.** The dimer used for primer extension was synthesised and purified as previously reported[67]. Specifically, cytidine 5′-monophosphate (0.5 mmol) was dissolved in dimethyl sulfoxide and reacted with 2-aminoimidazole hydrochloride (0.23 mmol) under vigorous stirring. Triphenylphosphine (5 mmol), 2,2′-dipyridyl disulfide (5 mmol), and triethylamine (10 mmol) were then added sequentially to the solution, which was stirred for 30 min. A sample was taken for [31]P NMR analysis to monitor reaction progress. The reaction mixture was subsequently precipitated by adding it to pre-cooled acetone (250 mL), diethyl ether (250 mL), and sodium perchlorate (saturated in acetone), followed by centrifugation and washing of the resulting solid with acetone and diethyl ether. The pellet was dried under a high vacuum to remove residual solvent.

**Oligonucleotide solid-phase synthesis.** DNA and RNA oligonucleotides were assembled using standard reagents and standard manufacturer protocols on a 1 μmol scale. DMTr-removal reagent consisted of 3% trichloroacetic acid in dichloromethane, the activator consisted of 0.25 M 5-ethylthio tetrazole in acetonitrile, the oxidiser consisted of a 0.02 M solution of iodine in pyridine:water:tetrahydrofuran (8:16:76 volume ratio), and the capping reagents consisted of (Cap A) a solution of acetic anhydride:pyridine:tetrahydrofuran (10:10:80 volume ratio)

and (Cap B) a 10% (v/v) solution of *N*-methylimidazole in tetra-hydrofuran. All oligonucleotides were deprotected from the solid support using 25% ammonium hydroxide:ethanol 4:1 volume ratio (1 mL total volume) for 17 h at 55 °C and concentrated in a Savant SC 110 A SpeedVac® Plus to a pellet. Oligonucleotides were then purified by ion exchange chromatography.

RNA oligomers were desilylated in DMSO:triethylamine trihydro-fluoride 2:3 volume ratio (100 μL:150 μL) for 2 h at 65 °C and then precipitated in cooled 1-butanol for 1 hour. Upon centrifugation, the pellet was recovered, and the supernatant was discarded. The pellet was further washed with 200 μL of 1-butanol. Deprotected oligonu-cleotides were purified by Strong Anion-Exchange (SAX) HPLC with solvent A (50 mM Tris buffer pH 7.6, 10% v/v MeCN) and solvent B (50 mM Tris pH 7.6, 1 M NaCl, 10% v/v MeCN), with a standard gradient of 0-75% over 15 min. Purified samples were desalted using Sep-Pak C18 Classic Cartridge (WaterTM). The Sep-Pak C18 cartridge was condi-tioned with 10 mL of MeCN, 10 mL of MeCN:water 1:1 volume ratio and 10 mL of 100 mM pH 7 NaOAc. The purified oligo was diluted to at least 2% v/v MeCN (1:4 dilution with water) and flowed through the cartridge at least twice for column loading. The bound oligonucleotide was washed with water (~25 mL), eluted from the column with 4 mL of MeCN:water 1:1 volume ratio, and concentrated into a pellet using a DNA concentrator.

**Determination of the critical salt concentration (CSC).** The robust-ness of complex coacervates is commonly assessed by their stability to salt, typically NaCl. The critical salt concentration corresponds to the highest NaCl concentration tolerated before the complete dissolution of coacervates. Turbidity was indirectly measured on a plate reader, reading the absorbance at 600 nm and using the relation:

$$\text{Turbidity} = 100 - \text{Transmittance}_{\%} = 100(1 - 10^{\text{Abs}_{\text{blank}} - \text{Abs}})$$

Samples of 100 μL (or 20 μL in the case of peptide/RNA mixtures) were prepared in 96-well plates (or 384-well plates) and titrated with concentrated stocks of NaCl (1, 3 or 5 M). The concentration of the salt stock was chosen to minimise the dilution of the sample during titra-tions (20% maximum dilution) and maximise the number of points measured during the steep decay of absorbance. At the end of the titration, all mixtures reached the turbidity of the blank (100 μL of MilliQ). The titration curves have a sigmoidal shape, and the CSC was calculated as follows: (i) the exact concentration of NaCl was calcu-lated at each point, taking into account the total volume in the well; (ii) the curve (turbidity *vs* NaCl concentration) was fitted every three points with a linear equation; (iii) the linear fit with the highest linear coefficient (absolute value) was used to identify the tangent at the inflection point ($y = ax + b$). The CSC was thus calculated as $CSC = -\frac{b}{a}$.

**Coacervation onset.** We define the coacervation onset as the amino acid concentration required for each peptide to form coacervates in the presence of oligonucleotides, assessed by turbidity measurements. Turbidity measurements were performed by monitoring absorbance at 600 nm in a plate reader upon titration of the oligonucleotide solution with a concentrated peptide stock until absorbance reached its maximum. As previously discussed, absorption was converted to turbidity, and the onset concentration corresponds to the amino acid concentration for turbidity >20%.

**Minimal complex concentration for coacervation.** We define the minimal complex concentration as the minimal concentration of peptide:oligonucleotide 4:1 concentration ratio required for coa-cervation. Coacervates were prepared as in previous experiments (20 mM amino acid concentration and 5 mM nucleotide concentra-tion, 20 μL samples), then serially diluted in a 384-well plate. Absor-bance at 600 nm was converted to turbidity, and the minimal complex

concentration for peptide/oligonucleotide mixtures was determined as the intercept between the x-axis and the tangent to the inflection point of the sigmoidal curve.

**Temperature stability with hot stage epifluorescence microscopy.** Borosilicate glass capillaries (internal section of 2 × 0.2 mm) were passivated using the same protocol as the coverslips. One capillary end was sealed with optical glue and cured under UV light (λ = 365 nm) for 5 min. Peptide/oligonucleotide mixtures containing 1% of Cy3-(TGAC)$_2$ were introduced in the capillary (approx. 30 μL), which was then completely sealed with a two-component epoxy resin and hardener glue. Glass capillaries were placed on a coverslip and subsequently on a copper plate connected to a Peltier element, enabling fine control over temperature.

**Fluorescence Recovery After Photobleaching (FRAP).** Cy3-labelled DNA oligonucleotides (Cy3-(TGAC)$_2$, Cy3-(TGAC)$_4$ and Cy3-(TGAC)$_8$, labelled on the 5′) were chosen as FRAP probes for peptide/oligonu-cleotide coacervates. Coacervates were prepared as described pre-viously, with the peptide added last to the microtube. Imaging was done ca. 30 min after sample preparation and placement in the observation chamber.

For each measurement, a droplet was chosen in the centre of the field of view (512 × 512 px) and imaged for 10 frames (every 1.117 s). A circular region of interest (ROI), selected inside the droplet (smaller than the droplet) before the acquisition, was bleached using the 633 nm laser line at 100% intensity. Post-bleaching images were col-lected at the same framerate until ROI intensity reached a plateau, which for our samples varied between 30–250 s (all profiles available in SI). Pre- and post-bleaching imaging was performed using the 633 nm laser line at 4–6% intensity and pinhole size set to 1 AU. A standard photomultiplier tube was used as a detector (480–720 nm). Three droplets in different FOVs were bleached for each sample, and the recovery curves were averaged. The fluorescence intensity pre- and post-bleaching was recorded and normalised to the average intensity pre-bleaching.

**Partition coefficients.** Partitioning of fluorescent client molecules was quantified using the equation[68]

$$K_p = \frac{I_{\text{droplet}} - I_{\text{dark}}}{I_{\text{dilute phase}} - I_{\text{dark}}}$$

The fluorescence intensity inside the droplet, $I_{\text{droplet}}$, was aver-aged among all droplets in the field of view (FOV) using a particle analysis plugin from ImageJ and a low threshold to prevent under-estimation. The intensity of the dilute phase was averaged for the entire FOV after droplets were removed. $I_{\text{dark}}$ corresponds to the intensity measured in a sample lacking any fluorophore at the same laser power used for the respective sample. Client molecules used include: Cy3-(TGAC)$_2$, Cy3-(ACTG)$_2$, FITC-r(ACUG)$_2$, Cy3-dA$_{11}$ and Magnesium Green.

**Broccoli aptamer reconstitution.** A minimal version of the Broccoli aptamer was split in strand A (5′-r(GCGGAGACGGUCGGGUCCAGAUA), 23nt) and strand B (5′-r(UAUCUGUCGAGUAGAGUGUGGGCUCCGC), 27nt) and its reconstitution was followed by DFHBI fluorescence in the presence of KCl. A 2000× DFHBI stock was prepared in DMSO and diluted 100× in 25 mM HEPES buffer before being added to the sample. The samples were prepared to ensure that coacervation takes place after aptamer reconstitution by mixing in the following order (unless otherwise stated): MilliQ, 25 mM HEPES buffer, 10 mM KCl), DNA$_8$/RNA$_8$/DNA$_{16}$ (5 mM nt), strand A (10 μM), strand B (10 μM), 5 mM DFHBI and peptide (20 mM amino acid). Measurements were performed before and after adding the peptide; fluorescence was recorded every

15 min for 1 h. For microscopy, coacervates containing the Broccoli aptamer and DFHBI were prepared and left to incubate for 30 min in sealed microscopy chambers.

**Primer extension reaction.** Primer extension reactions in the presence of coacervates were performed with 25 mM HEPES (pH 8.0), $R_4$ or $R_6$ 20–40 mM (amino acid concentration), nucleic acid host strand 5-10 mM (nucleotide concentration), 3 μM 6-FAM-labelled primer and 4 μM template. Activated dimer stocks were resuspended in water and HEPES buffer pH 8.0 (final concentration of 50 mM), resulting in a stock concentration of 100 mM. The required amount for the host strand was dried in a centrifuge tube using a DNA concentrator, followed by resuspension in water (5 μL).

First, water was added to 25 mM HEPES buffer (pH 8.0), followed by peptide, and premixed primer-template duplex. The resuspended host oligonucleotide strand (5 μL) was then added, resulting in the formation of coacervates (as noticed by increased turbidity). The reaction was initiated by the addition of activated CC dimer (2.5 mM final concentration), finally followed by $MgCl_2$ (5 mM final concentration). The reaction was mixed vigorously and quenched (4 μL) at the various time points (1 h, 3 h, 6 h, 24 h); only 2 μL were taken at the 0 h time point. Timepoints were quenched using 8 M urea (44 μL) containing 72 mM NaCl.

Control primer extension experiments were repeated as described above but with NaCl (1 M) added to dissolve the coacervates. Aliquots for control experiments containing NaCl were quenched using 8 M urea (36 μL). Additional control experiments were repeated without the presence of peptide, with and without 1 M NaCl. Note that all volumes were adjusted to a final volume of 20 μL by varying the amount of water added.

All primer extension species were resolved by 20% (19:1) denaturing PAGE with 8 M urea using 3 μL of the quenched aliquots. Polyacrylamide gels were cast using 20% (acrylamide:bis-acrylamide 19:1) denaturing gels (8 M urea) and run in 1× TBE buffer (89 mM Tris, 89 mM boric acid, and 2 mM EDTA) in a 16 cm wide × 14 cm long × 0.8 cm thick gel. Gels were pre-run at 10 W for at least 15 min before loading. Gels were initially run at 4 W until the markers (95% formamide, 0.025% bromophenol blue, 0.025% xylene cyanol) had loaded onto the gel matrix and separated. The cell voltage was then increased to 10 W and run for at least two hours. Gels were imaged on an Amersham TYPHOON. All band intensities were quantified using the ImageQuantTM software (using the background reduction function and manual band detection).

## Computational methods

**Atomistic force-field simulations.** The simulations were performed using the Amber14SB force field for peptides[69], OL3 parameters for RNA[70], and bsc1 for DNA[71]. All systems were solvated using the TIP4P-FB water model[72], and compatible ion parameters were applied for sodium ($Na^+$) and chloride ($Cl^-$) ions.

Temperature control was managed by a Langevin thermostat, set to 298 K with a friction coefficient of 1 ps$^{-1}$. For simulations conducted in the isothermal-isobaric (NPT) ensemble, the pressure was maintained at 1 atmosphere using a Monte Carlo barostat[73] with updates applied every 25 steps.

Using the LFMiddle discretisation scheme[74], the Langevin integrator was employed to propagate the system dynamics. Hydrogen mass repartitioning was used, enabling a time step of 4 fs during the production simulations, which was further supported by constraining all bonds involving hydrogen atoms using the CCMA algorithm[75]. Non-bonded interactions were computed with a cutoff distance of 0.9 nm. Long-range electrostatics were handled using the Particle Mesh Ewald (PME) method[76]. All simulations were performed using OpenMM 8.1.2[77], leveraging the CUDA platform in mixed precision mode. Energy minimisation was performed using OpenMM's built-in local energy minimiser, which utilises the L-BFGS optimisation algorithm[78] until it converged to a tolerance of 10 kJ mol$^{-1}$ nanometer$^{-1}$.

**Monomer preparation.** Initial monomer structures were built using PyMOL 2.5.7[79]. Single-stranded RNA and DNA 8-mers were constructed in an extended conformation approximating B-form dihedral angles. Peptides composed of polyarginine were prepared in an extended conformation. Peptides were modelled with protonated N-termini ($NH_3^+$) and deprotonated C-termini ($COO^-$) to reflect physiological conditions. Nucleotides were prepared without the 3′ phosphate group to match experimental conditions. Each monomer's initial configuration was solvated in a cubic box with a minimum of 5 Å between any solute atom and the box edge. The systems were neutralised, and ionic strength adjusted to 30 mM NaCl. Energy minimisation was performed, followed by a 100 ns NVT equilibration at 298 K. Evenly spaced configurations from the last 80 ns of the monomer simulations were extracted to build multi-chain systems.

**Multi-chain system preparation.** Multi-chain systems were constructed by placing monomers using Packmol 20.14.4[80], enforcing a minimum distance of 10.0 Å between any two atoms to prevent overlaps. Each system consisted of 8 nucleotides (either RNA or DNA) and 36 polyarginine peptides, with one nucleotide and four peptides placed randomly within each octant of a cubic box with a side length of 140.0 Å. The assembled systems were solvated in a cubic box 145.0 Å per side, ensuring a minimum of 5 Å between any solute atom and the box edge under periodic boundary conditions, before being neutralised and brought to an ionic strength of 100 mM NaCl.

**Equilibration protocol.** After minimisation, the systems were relaxed through the following steps:

- 250 ps of NPT simulation at 298 K and 1 atm with heavy atom positional restraints of 15 kcal mol$^{-1}$ Å$^{-2}$ applied to all peptide and nucleotide heavy atoms, using a 2 fs time step.
- 250 ps of unrestrained NPT simulation at 298 K and 1 atm, with a 2 fs time step.
- 500 ps of unrestrained NVT simulation using a 2 fs time step.

**Production simulations.** Production simulations were carried out in the NVT ensemble for 800 ns per replicate, with 5 independent replicates for each system, totalling 4.0 μs of simulation time per system. Each replicate began from an independently prepared configuration and used different random number seeds to ensure statistical independence.

Trajectory frames were saved every 0.8 ns. The last 400 ns of each simulation (corresponding to 500 frames) were used for contact analysis. Contacts between molecules were defined based on a cutoff distance of 0.45 nm between heavy atoms and were analysed using a custom Python script utilising CuPy.

**Interaction analysis.** Trajectories were analysed using the Python package MDTraj[81]. Hydrogen bonds were assessed using the Wernet-Nillson criteria[82]. Ionic interactions were defined as occurring when the CZ atom from an arginine sidechain approaches OP1 or OP2 atoms from a phosphate group closer than 0.6 nm without a hydrogen bond being established between the two residues. Following previous work[42], Arg-nucleobase stacking interactions were defined as occurring when the CZ atom from an arginine sidechain approaches the centre-of-geometry of a nucleobase ring with the angle between the planes of the guanidium group and the nucleobase ring less than 30°.

**Coarse grained model.** Coarse grained simulations were carried out using an OpenMM implementation of the newly released residue level coarse-grained model Mpipi-Recharged, which is designed to accurately model the liquid-liquid phase separation of highly charged

biomolecular condensates[83]. The model employs a one-bead-per-residue representation for amino acids and unstructured single-stranded RNA. Parameters are available for all 20 natural amino acids and uridine in RNA. Full details of the model and its parameters are available in the recent publication.

**Coarse grained simulation procedure.** Using the Mpipi-Recharged model, direct coexistence simulations[84] of various mixtures of polyR peptide and polyU RNA were performed. For each system we simulated, initial extended configurations of monomers were prepared and relaxed for 10 ns at 300 K. Following this, copies of the relaxed monomer were placed into a rectangular box on a regular grid pattern. The periodic boxes used were ~144 nm × 24 nm × 24 nm and periodic boundary conditions were applied in all 3 directions. An OpenMM Custom External Force was then used to pull all monomers to the centre of the box. After 10 ns of pulling, a dense slab formed at the centre of the box, at which point the pulling force was switched off. Subsequently, the systems were simulated for 1 ms without the external force to allow the formation of coexisting high- and low-density phases. All multichain coarse-grained simulations were carried out at a temperature of 300 K and an implicit salt concentration of 100 mM NaCl. Integration was carried out using a Langevin Middle Integrator with timestep of 10 fs and collision frequency of 0.01/ps.

**Coarse grained simulation analysis.** Custom Python scripts utilising MDTraj and CuPy were used to analyse coarse grained simulations[85]. Contacts were defined as occurring when two coarse grained beads $i$ and $j$ are closer than 0.5 $(\sigma i + \sigma j)$ +0.1 nm, where $\sigma i$ is the characteristic length scale associated with bead $i$.

### Reporting summary
Further information on research design is available in the Nature Portfolio Reporting Summary linked to this article.

## Data availability
Raw data used in this study are available in the Apollo – University of Cambridge repository under accession code: https://doi.org/10.17863/CAM.120235. Source data are provided in the Source Data file.

## Code availability
Code used in all simulations is available, with examples, at: https://github.com/orgs/CollepardoLab/MinimalCoacervates.

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

## Acknowledgements

We thank Jean-Daniel Fauny and Romain Vauchelles (IBMC, Unistra) for access to the confocal microscopy facility, Valentin Bauer (ISIS, Unistra) for support with SPPS and peptide purification, and Tiemei Lu for input on the initial experimental setup. We thank Prof. Pietro Cicuta and Dr Jurij Kotar (Cavendish Laboratory) for support to obtain a dataset on the temperature stability of minimal coacervates. We also thank Dr David A. Russell (University of Cambridge) for insightful comments on the manuscript and relevant literature provided. We further thank Dr Vikki Cantrill for her support in editing this manuscript.The authors acknowledge funding from the NWO (Dutch Research Council) via a Rubicon Fellowship (019.222EN.011 to K.K.N.), the Human Frontier Science Program Organization (HFSPO) via an Early Career Research Grant (RGY00062/2022, to C.B. and D.K.O.), the ERC (Starting Grant) under the European Union's Horizon Europe research and innovation programme (GA 101162933 to C.B.), the Federation of European Biochemical Society via a FEBS Excellence Award (to C.B.), the Agence Nationale de la Recherche via an ANR AAPG JCJC 2022 (to C.B.), the CSC Graduate School funded by the Agence Nationale de la Recherche (CSC-IGS ANR-17-EURE-0016 for doctoral funding to F.R.), the University of Strasbourg Institute for Advanced Study (USIAS) via a USIAS Fellowship (to C.B.), the Foundation Jean-Marie Lehn, the Biotechnology and Biological Sciences Research Council via a BBSRC Discovery Fellowship (BB/X010228/1 to R.R.S.), the UKRI EPSRC under the UK Government's guarantee scheme (EP/Z002028/1 to R.C.G.), following funding by the ERC (Consolidator Grant) under the European Union's Horizon Europe research and innovation programme, the Winton Programme for Physics of Sustainability (for doctoral funding to M.J.M.), the NSERC via a NSERC Discovery Grant (RGPIN 2020-05043 to D.K.O.) and a NSERC Alliance Catalyst Grant (ALLRP 57555822 to D.K.O.). This project made use of time on HPC granted via the UK High-End Computing Consortium for Biomolecular Simulation, HECBioSim (http://hecbiosim.ac.uk), supported by EPSRC (EP/R029407/1 to R.C.G.).

## Author contributions

K.K.N., F.Z.M. and C.B. designed and conceptualised the study. K.K.N. and F.Z.M. performed all experiments aimed at the systematic characterisation of primitive coacervates. F.R. performed peptide synthesis and characterised the peptide material. C.B. supervised peptide synthesis and coacervate experiments. K.O.R. conceptualised and performed simulations. J.H., M.J.M. and K.O.R. analysed simulation data. R.C.G. supervised simulation work. J.S.S. and J.D.R. prepared nucleic acid material, activated dinucleotides and conducted all primer extension experiments. D.K.O. supervised primer extension experiments and analysed primer extension PAGE data. F.Z.M., K.K.N. and R.R.S. performed the FRAP experiments and analysed the data. K.K.N., R.R.S., R.C.G., D.O.F. and C.B. acquired funding. All authors discussed the results, contributed to the writing of the first draft of the manuscript, provided comments and approved the final version of the manuscript.

## Competing interests

The authors declare no competing interests.
