## [Transparent Peer Review file · Nature Communications]

Differential stability and dynamics of DNA-based and RNA-based coacervates affect non-enzymatic RNA chemistry

Corresponding Author: Dr Claudia Bonfio

Version 0:

Reviewer comments:

Reviewer #1

(Remarks to the Author)

The authors present an exhaustive set of experiments mapping the coacervate properties based on short—prebiotically plausible—oligonucleotides with model polyR peptides. They report that the presence of DNA oligonucleotides (which they argue may better reflect prebiotic scenarios) in the coacervates increases their fluidity and facilitates better diffusion of oligonucleotides. This is a major and novel finding of the manuscript and, in my opinion, deserves the attention of the broader scientific community.

I read the paper with interest and really enjoyed the new findings. However, the content was not always easy to follow, and this could likely be improved by more thorough work on figure legends and referencing—especially to the extensive supplementary material (please see minor points below). My major comments concern the choice of peptide in the studied system and the conclusions drawn from that, especially in the context of recent work in the field, as listed below.

Major comments:

1. Peptide choice and relevance:

While I understand the choice of polyR as a model peptide, I find this to be a major weakness of the study if the main aim is to investigate prebiotically plausible coacervation. Peptide synthesis was likely not templated in prebiotic environments, and the occurrence of Arg was probably quite sparse. Moreover, recent studies have shown that heteropeptides (where cationic and neutral amino acids are interspersed) lead to coacervates with higher ribozyme activity (Ref 17) and may positively affect Mg^{2+} partitioning. When homopeptides have been used, poly-Lys sequences have been shown to better preserve ribozyme activity (Ref 18; see also LeVay et al., 2021). I believe the authors should comment on these findings more thoroughly in the discussion, as some of the peptide-related properties may have similar effects as heterogeneity on the nucleotide side. This also relates to the following points:

a. Lines 346–348: The suggestion that “non-enzymatic RNA polymerization may have been inefficient in crowded prebiotic settings in which reactive peptides were abundant” is far-fetched and contrasts, for example, with Ref 12. If the authors wish to raise this point, I’m afraid they would need to conduct additional experiments (e.g., using polyK or peptides that more accurately reflect plausible prebiotic scenarios).

Additionally, I found lines 343–346 difficult to follow in light of the data presented in Figs. S29 and S30. Could the authors please verify that the data is described correctly, and perhaps also reference Table S14 (if that is the source of the data shown in those figures)? I apologize if I missed something there.

b. Lines 350–359: It is proposed that the reaction takes place in both the coacervate and dilute phases. Did the authors perform any experiments differentiating ribozyme occurrence in these two phases? Have they considered the impact of peptides on Mg^{2+} partitioning and the subsequent effect on activity? (As observed previously in Ref. 17, especially for homopeptides.)

2. Consistency issues:

- a. Can the authors explain why R4 was used for microscopy experiments monitoring coacervation with the ribozyme, while R6 was used for the PE efficiency measurements (Fig. 4a–b vs. 4e)? Apologies if I missed this explanation somewhere.
 - b. Fig. 1 – Why was thermal stability tested only for R/RNA and R/DNA coacervates, and not R/E?
3. Line 48 (Introduction): The authors acknowledge previous work on peptide–ribozyme systems in coacervates but refer only to “homopeptides.” However, some of the cited papers (e.g., Ref 17) also used heteropeptides and demonstrated their advantages in supporting ribozyme function over homopeptides. This should be corrected (e.g., by deleting “homo-” or providing a more detailed description), and the findings should be discussed in light of their implications for the system characterized in this study.
4. DNA and RNA oligonucleotides as “heterogeneous mixtures”: The authors present the DNA and RNA oligonucleotides as heterogeneous mixtures of prebiotic oligonucleotides. I would advise more caution here, as in both RNA and DNA cases, specific sequences of ACUG/ACTG (or their repetitions) are used. There are studies highlighting the importance of sequence structure content in coacervation. Without testing, for example, shuffled variants of the oligos, the results may be biased and do not truly reflect “heterogeneous mixtures.”

Minor issues:

1. Graphical abstract: The arrows pointing to peptide/RNA/DNA also point to their respective monomeric units (amino acids/nucleotides). I think this should be avoided to prevent confusion.
2. Statistical reporting: It would be helpful (and likely required by the journal) for each main figure to include information about the number of replicates performed and relevant statistical analyses.
3. Line 256: “are known inhibit” — likely missing a “to”; should be “are known to inhibit.”
4. Figure legends: Wherever data in tables correspond to data presented in figures, it would be helpful to indicate that connection in the figure legends.

Reviewer #2

(Remarks to the Author)

This work reports that short heterogeneous oligonucleotides (RNA, DNA, and their combinations) spontaneously undergo phase separation with short poly-arginine chains to generate so-called primitive coacervates, which likely impacted prebiotic RNA chemistry. This work challenged the notion that the emergence of coacervates succeeded in synthesizing long, sequence-specific, functional polymers (i.e., homopeptides and ribozymes). The authors conducted extensive experimental work to characterize the coacervates in terms of stability towards salt that affects the charge matching between the positive peptide chain and the negative RNA/DNA, the composition (peptide/nt ratio), mobility, and temperature. This was complemented by MD simulation, which gave a molecular-level account for the stability through counting the various types of interactions between the peptide and the RNA or DNA. The results showed that RNA provides stability to the coacervates, while adding DNA led to lower viscosity, facilitating the diffusion of molecules in and out of the coacervates. These findings suggest that compartmentalization via coacervation could have coincided with early stages of non-coded amino acid and nucleotide polymerization. This is a strong and significant aspect of the manuscript.

The weakness of the manuscript is that the authors do not present any evidence for a functionality that takes place within the coacervate, which otherwise would not take place as effectively as in solution, thus supporting their function as protocells. This limits the significance and impact of this work.

Major issues:

1. The sentence in p. 11 “These findings indicate that coacervates made of short DNA oligonucleotides and Arg peptides — characterised by weaker and less abundant interactions, and hence remarkably enhanced mobility, than their RNA-based counterparts (Fig. 2b) — enable more efficient nucleic acid folding” is not fully supported by the experiment. The fluorescence attenuation can be just a question of different viscosity, which affects the chance of the two parts of the Broccoli aptamer to find each other and fold properly. Indeed, it can also take place in low-viscosity coacervates, but folding is more effective in solution; there is no need for the coacervate to achieve folding.
2. Fig. S25 — I do not see the results for R10/E10. Why is there a reduction in the intensity of the control with time (Fig. S25B)? Is it just an experimental error, or is there some degradation? What does this mean in terms of the small increases observed for the coacervates? Why is the emission different in Fig. S25A and B? The same scale should be used for both panels.
3. The subtitle “Primitive coacervates enable RNA polymerization,” though correct, is misleading; polymerization is more effective in solution. The title seems to hint that the reaction occurs in the coacervates and renders some functionality to the coacervation. To my disappointment, the effect of the coacervation is to enable RNA polymerization in the dilute phase by sequestering peptides that would otherwise inhibit the reaction. While this is an interesting result, it does not support the idea that coacervates act as early cells, where one would expect some function to occur within the coacervate.
4. In Fig 4f there is a correlation between the PE efficiency and the viscosity of the coacervates as determined by FRAP,

showing that the efficiency is reduced as the viscosity increases and as a conclusion the authors write “ we reason that the coacervate phase, despite its minimal volume fraction significantly contributes to the efficiency of RNA polymerization” Nevertheless, there is no explanation as to how. Is it possible that the dilute phase viscosity also increases? It is related to the amount of peptide recruited; the more, the better. The authors could have labeled the peptide and followed its concentration within the coacervates. Do they have any evidence that the reaction also occurs within the coacervates? Minor issues.

Please add units to the x-axis of Fig. 4f

Figure 4A : for R4/DNA16, the rim of the droplets shows stronger fluorescence than the center, in contrast to the other contracts. Please explain.

Fig. 1d - The curve for RN/HNA8 is not linear.

Reviewer #3

(Remarks to the Author)

The study demonstrates that short, mixed-sequence oligonucleotides can form coacervates with tri- or dipeptides under conditions plausible for prebiotic chemistry, reflecting the complexity of early Earth environments. It finds that RNA-based coacervates are uniquely stable compared to those formed with DNA or peptides alone, challenging previous assumptions about the roles of DNA and peptides in early compartment formation and highlighting RNA's significance in prebiotic chemistry. Additionally, the research reveals that DNA unexpectedly enhances the diffusion of reactive oligonucleotides, which is important for non-enzymatic RNA polymerization. By systematically comparing different types of coacervates under identical conditions, this work redefines our understanding of early compartmentalization and supports a model for the co-emergence of nucleic acids and peptides, marking a significant advance in origins-of-life research.

The authors investigated RNA folding within coacervates by introducing the Broccoli aptamer as a model RNA. While this is a compelling approach to assess RNA structure in these primitive compartments, it would be valuable to discuss the degree of cooperativity between RNA folding and peptide interactions within the coacervate phase. The study shows that coacervates formed from short DNA and arginine peptides, which feature weaker and more dynamic interactions, better preserve the aptamer's folded structure compared to RNA-based coacervates, where stronger peptide-RNA interactions can partially hinder proper folding. This suggests that while coacervates can promote RNA folding by concentrating RNA and providing a unique microenvironment, the nature and strength of peptide-nucleic acid interactions are critical: excessive or overly strong interactions may reduce RNA mobility and folding efficiency, whereas more fluid, less interactive coacervates (such as those with DNA) support folding more cooperatively. To strengthen the manuscript, I recommend a more detailed and quantitative exploration of RNA-peptide cooperativity within coacervates, including additional structural and dynamic analyses.

Further discussion on how the balance between stability and fluidity in these compartments modulates RNA-peptide cooperativity would enhance our understanding of the functional potential of prebiotic coacervates.

Complement fluorescence data with additional structural techniques such as NMR or the molecular simulations, to directly probe RNA conformational states and dynamics within coacervates. This could clarify whether peptides stabilize native RNA folds or induce alternative structures

Version 1:

Reviewer comments:

Reviewer #1

(Remarks to the Author)

I am satisfied with the comments provided by the authors and the revised manuscript. Although more experimental results (mainly regarding prebiotically more plausible peptides and deciphering the relationship between the coacervate phase and the RNA processes) were not presented in the revised version, I value the currently presented original results and I think the manuscript merits its publication. I understand that the "missing pieces" will be a subject of a follow-up study.

Reviewer #2

(Remarks to the Author)

The authors have addressed my comments, clarified the points needed in the revised manuscript and I support publication of the manuscript as is.

We thank the Reviewers for their positive assessment of our manuscript, their detailed evaluation of our work, their comments, and their many helpful suggestions for improving the manuscript. Reviewers' comments are included in the sections below in black, and our response is given in blue.

Reviewer #1

The authors present an exhaustive set of experiments mapping the coacervate properties based on short—prebiotically plausible—oligonucleotides with model polyR peptides. They report that the presence of DNA oligonucleotides (which they argue may better reflect prebiotic scenarios) in the coacervates increases their fluidity and facilitates better diffusion of oligonucleotides. This is a major and novel finding of the manuscript and, in my opinion, deserves the attention of the broader scientific community.

I read the paper with interest and really enjoyed the new findings. However, the content was not always easy to follow, and this could likely be improved by more thorough work on figure legends and referencing—especially to the extensive supplementary material (please see minor points below). My major comments concern the choice of peptide in the studied system and the conclusions drawn from that, especially in the context of recent work in the field, as listed below.

We appreciate and share the Reviewer's enthusiasm for our work, and we thank them for their positive assessment of our manuscript. Following the Reviewer's comments, we adjusted and amended the main text, figures and SI.

Major comments:

1. Peptide choice and relevance:

While I understand the choice of polyR as a model peptide, I find this to be a major weakness of the study if the main aim is to investigate prebiotically plausible coacervation. Peptide synthesis was likely not templated in prebiotic environments, and the occurrence of Arg was probably quite sparse. Moreover, recent studies have shown that heteropeptides (where cationic and neutral amino acids are interspersed) lead to coacervates with higher ribozyme activity (Ref 17) and may positively affect Mg^{2+} partitioning. When homopeptides have been used, poly-Lys sequences have been shown to better preserve ribozyme activity (Ref 18; see also LeVay et al., 2021). I believe the authors should comment on these findings more thoroughly in the discussion, as some of the peptide-related properties may have similar effects as heterogeneity on the nucleotide side.

We thank the Reviewer for raising this point. As correctly pointed out by the Reviewer, a number of studies have already been published on coacervates composed of either DNA or RNA with polyR, polyK and heteropeptides (e.g., comprising RGG repeats) as models for primitive cells. Due to the lack of systematic studies on the potential differences in coacervation induced by RNA or DNA, in our manuscript we intentionally chose to focus on nucleic acids, specifically short heterogeneous oligomers of RNA and DNA.

We agree with the Reviewer that a more systematic exploration of the prebiotic peptide space is due (and it is the subject of a manuscript we are currently preparing for publication, the results of which are partially included in our most recent preprint). Still, we believe that the short peptides employed in our study (in particular, tetra- and tripeptides) represent a significant improvement in prebiotic plausibility due to the lower number of condensation events required for their formation, as suggested in Ref. 24,

compared to long (homo)peptides. Additionally, as a prebiotic route for arginine, but not lysine, has been published (Ref. 3), we chose to use Arg-based peptides in this study.

In the Results section of the main text, we have now commented on this aspect.

This also relates to the following points:

a. Lines 346–348: The suggestion that “non-enzymatic RNA polymerization may have been inefficient in crowded prebiotic settings in which reactive peptides were abundant” is far-fetched and contrasts, for example, with Ref 12. If the authors wish to raise this point, I’m afraid they would need to conduct additional experiments (e.g., using polyK or peptides that more accurately reflect plausible prebiotic scenarios).

We apologise for the lack of clarity in our sentence. In our statement, we refer to non-enzymatic template-directed RNA elongation rather than ribozyme-driven RNA polymerisation, as studied in Refs. 12 and 17. Specifically, we observed that even a simple amino acid such as glycine negatively affects the stability of the activated imidazolium-bridged dimer (Fig. S28), causing a lower efficiency of RNA elongation. Similarly, Figs. S29 and S30 show how the presence of Arg-based peptides partially inhibits template-directed RNA elongation. The reactivity between amino acids and activated phosphates has been also observed in studies focused on phosphoramidate bond formation:

- Radakovic, *Biochemistry* 2021: <https://pubs.acs.org/doi/10.1021/acs.biochem.0c00943>

- Roberts, *JACS* 2022: <https://pubs.acs.org/doi/10.1021/jacs.2c00772>

To address the point raised by the Reviewer, we have now rephrased our statement in the main text as follows:

“... non-enzymatic template-directed RNA elongation may have been inefficient in crowded prebiotic settings in which reactive peptides, prone to degrade the activated dimer, were abundant...”

Additionally, I found lines 343–346 difficult to follow in light of the data presented in Figs. S29 and S30. Could the authors please verify that the data is described correctly, and perhaps also reference Table S14 (if that is the source of the data shown in those figures)? I apologize if I missed something there.

We apologise for the lack of clarity. We have added the explicit cross-reference between Table S14 and Figs S29, S30, 4d and 4e. We have also included the two references mentioned above to the references list in the main text. We have finally rephrased the main text as follows:

“Due to the reactivity of amines towards the activated dimer (Fig. S28), we expected Arg peptides to compromise primer extension. We thus evaluated the efficiency of primer extension in the presence of R₆, adding NaCl to prevent phase separation. Regardless of the concentration of R₆, primer extension was 1.5 times less efficient in the presence of peptide than without after 24 hours (Table S14 and Figs. S29 and S30).”

b. Lines 350–359: It is proposed that the reaction takes place in both the coacervate and dilute phases. Did the authors perform any experiments differentiating ribozyme occurrence in these two phases? Have they

considered the impact of peptides on Mg^{2+} partitioning and the subsequent effect on activity? (As observed previously in Ref. 17, especially for homopeptides.)

As the Reviewer mentions, we measured the efficiency in primer extension without differentiating between the coacervate and the dilute phases. As the volume of the coacervate phase is very small (about 1% of the total mixture), likely due to the reduced length and concentration of the peptides and oligonucleotides involved, it was not always possible to fully isolate the coacervate phase.

Work performed in Ref. 17 shows that peptide-RNA interactions drive the depletion of Mg^{2+} from coacervates. We attempted to quantify Mg^{2+} partitioning in coacervates using Magnesium Green as a probe, which fluoresces upon Mg^{2+} binding (as shown below, scale bar: 10 μm). While we included the Mg^{2+} partitioning data in the SI (Table S11), we are cautious in asserting that the primer extension reaction takes place only in the dilute phase, as this probe might not be suitable for coacervates.

In contrast, the presence of excess peptide in the dilute phase (Figs. 2e and S18), which inhibits the primer extension (see comment above), might suggest that the reaction takes place, at least partially, in the coacervate phase. Besides, previous studies (Ref. 18) showed that, in the presence of coacervates, primer extension is relatively insensitive to magnesium concentration.

Overall, our work demonstrates that non-enzymatic template-direct RNA elongation is compatible with primitive coacervates, regardless of where the reaction takes place, and that both primer and elongated product are present in the droplets. Reviewer 1 might also find interesting our comment to a similar concern raised by Reviewer 2 (*vide infra*).

2. Consistency issues:

a. Can the authors explain why R4 was used for microscopy experiments monitoring coacervation with the ribozyme, while R6 was used for the PE efficiency measurements (Fig. 4a–b vs. 4e)? Apologies if I missed this explanation somewhere.

We apologise for the lack of clarity related to Fig. 4. Panels a-b refer to the encapsulation of an RNA aptamer, while panels c-f refer to the efficiency of non-enzymatic template-directed RNA elongation. The RNA aptamer and the RNA oligonucleotides employed for primer extension are not related, and we did not perform experiments using ribozymes.

We performed the primer extension reaction with both R4 and R6, and compared them in Fig. S36. However, as the coacervate volume typically correlates to the size of its polymeric components, and R4-

based coacervates have lower critical salt concentrations than R_6 -based analogues, employing R_6 instead of R_4 led to the preservation of a higher volume of coacervate phase, and thus an enhanced reproducibility and efficiency of the primer extension reaction. Yet, we observed the same trends in the series of experiments with R_4 (Table S14 and Fig. S36) and R_6 (Fig. 4d-f) for primer extension reactions.

b. Fig. 1 – Why was thermal stability tested only for R/RNA and R/DNA coacervates, and not R/E?

We thank the Reviewer for identifying this missing experiment. We initially did not perform this experiment, as we focused only on the difference between DNA and RNA coacervates. However, we recognise the importance of comparing nucleic acid-based coacervates with peptide-peptide coacervates. As such, we performed the experiment requested by the Reviewer (Fig. S8b) and we modified the main text as follows:

“A similar thermal stability was observed for peptide/peptide coacervates only when longer polymers were employed (R_{10}/E_{10}) (Fig. S8).”

3. Line 48 (Introduction): The authors acknowledge previous work on peptide–ribozyme systems in coacervates but refer only to “homopeptides.” However, some of the cited papers (e.g., Ref 17) also used heteropeptides and demonstrated their advantages in supporting ribozyme function over homopeptides. This should be corrected (e.g., by deleting “homo-” or providing a more detailed description), and the findings should be discussed in light of their implications for the system characterized in this study.

Noted and corrected as follows:

“Interestingly, it was recently reported that coacervates comprising heteropeptides with low charge density enhance ribozyme mobility and maximise Mg^{2+} uptake compared to coacervates composed of polyarginines.¹⁷”

4. DNA and RNA oligonucleotides as "heterogeneous mixtures": The authors present the DNA and RNA oligonucleotides as heterogeneous mixtures of prebiotic oligonucleotides. I would advise more caution here, as in both RNA and DNA cases, specific sequences of ACUG/ACTG (or their repetitions) are used. There are studies highlighting the importance of sequence structure content in coacervation. Without testing, for example, shuffled variants of the oligos, the results may be biased and do not truly reflect "heterogeneous mixtures."

We thank the Reviewer for their comment. We chose ACUG- and ACTG-based sequences as models for heterogeneous oligonucleotide mixtures because each sequence includes an equal amount of each nucleobase and does not fold in any secondary structure. We agree with the Reviewer that shuffled variants of the oligonucleotides, and even libraries of oligonucleotides, could have further enhanced the prebiotic plausibility of our system. We have now included an explanation for our choice of oligonucleotides in the main text as follows:

“Additionally, we chose oligonucleotide sequences with an ACTG/ACUG motif to avoid the formation of secondary structures and minimise nucleobase/sequence biases.”

Minor issues:

1. Graphical abstract: The arrows pointing to peptide/RNA/DNA also point to their respective monomeric units (amino acids/nucleotides). I think this should be avoided to prevent confusion.

Noted and corrected.

2. Statistical reporting: It would be helpful (and likely required by the journal) for each main figure to include information about the number of replicates performed and relevant statistical analyses.

Noted and corrected.

3. Line 256: “are known inhibit” — likely missing a “to”; should be “are known to inhibit.”

Noted and corrected.

4. Figure legends: Wherever data in tables correspond to data presented in figures, it would be helpful to indicate that connection in the figure legends.

Noted and corrected.

Reviewer #2 (Remarks to the Author)

This work reports that short heterogeneous oligonucleotides (RNA, DNA, and their combinations) spontaneously undergo phase separation with short poly-arginine chains to generate so-called primitive coacervates, which likely impacted prebiotic RNA chemistry. This work challenged the notion that the emergence of coacervates succeeded in synthesizing long, sequence-specific, functional polymers (i.e., homopeptides and ribozymes). The authors conducted extensive experimental work to characterize the coacervates in terms of stability towards salt that affects the charge matching between the positive peptide chain and the negative RNA/DNA, the composition (peptide/nt ratio), mobility, and temperature. This was complemented by MD simulation, which gave a molecular-level account for the stability through counting the various types of interactions between the peptide and the RNA or DNA. The results showed that RNA provides stability to the coacervates, while adding DNA led to lower viscosity, facilitating the diffusion of molecules in and out of the coacervates. These findings suggest that compartmentalization via coacervation could have coincided with early stages of non-coded amino acid and nucleotide polymerization. This is a strong and significant aspect of the manuscript.

We thank the Reviewer for their positive evaluation of our work.

The weakness of the manuscript is that the authors do not present any evidence for a functionality that takes place within the coacervate, which otherwise would not take place as effectively as in solution, thus supporting their function as protocells. This limits the significance and impact of this work.

As mentioned above, we aimed at demonstrating the compatibility between RNA primer extension and primitive coacervates, and thus measured the efficiency in primer extension without differentiating between the coacervate and the dilute phases. As the volume of the coacervate phase is very small (see also comment above), it was not possible to monitor the reaction separately in each phase. Moreover, it is often the case that in a biphasic system, the reaction takes place, to some extent, in both phases. However, if the reaction did not at least partially take place in the coacervate phase, the composition of the coacervate phase would not influence the efficiency of the primer extension reaction as instead we observe.

To respond to the Reviewer's comment, we estimated the yield of the elongated product over time in a single-phase system in the absence and presence of inhibitors (e.g., reactive nucleophiles inducing the degradation of the activated dimer), as well as in a two-phase system with different viscosities. We broke down the process of primer extension into four reaction steps: (i) primer-template binding, (ii) primer-activated dimer reaction, (iii) dissociation of elongated primer and template, and (iv) inhibitory reaction between dimers and nucleophiles (e.g., R_4). We used the partitioning coefficients measured in our work, or reasonable approximations thereof, and assumed that the rate of transfer between the phases is not limiting.

In a single-phase system, we observe that peptide inhibition decreases the yield of the extended primer, as also shown in Fig. S28. In a two-phase system (with 1% coacervate phase), the measured yield of primer extension after 24 h is recovered (up to 85% for the parameters tested), regardless of whether the reaction fully takes place in the coacervate phase or not. If we then assume that, in viscous coacervates, all bimolecular rate constants are lowered by a factor of 2 in comparison to the dilute phase, we still observe

higher yields in a two-phase system than in the single-phase condition, although to a lesser extent (60%). The change in viscosity in this model calculation is to account for the effect we observe in DNA vs. RNA coacervates.

As such, there is no evidence suggesting that the reaction does not take place, at least partially, in the coacervate phase, since, despite the small coacervate volume, a significant fraction of template and product partitions in the droplets (Table S11).

Major issues:

1. The sentence in p. 11 “These findings indicate that coacervates made of short DNA oligonucleotides and Arg peptides — characterised by weaker and less abundant interactions, and hence remarkably enhanced mobility, than their RNA-based counterparts (Fig. 2b) — enable more efficient nucleic acid folding” is not fully supported by the experiment. The fluorescence attenuation can be just a question of different viscosity, which affects the chance of the two parts of the Broccoli aptamer to find each other and fold properly. Indeed, it can also take place in low-viscosity coacervates, but folding is more effective in solution; there is no need for the coacervate to achieve folding.

We apologise for the lack of clarity in our sentence. As mentioned in the SI, the Broccoli aptamer is pre-assembled in solution before the addition of coacervates. As such, our aim is not to enable RNA folding within coacervates, but rather to confirm that coacervates can maintain RNA secondary structures. Still, we agree with the Reviewer that our results strongly suggest, but do not definitively indicate, that primitive DNA-based coacervates enable the retention of folding more efficiently. We have now rephrased the sentence in the main text as follows:

“These findings indicate that coacervates made of short DNA oligonucleotides and Arg peptides — characterised by weaker and less abundant interactions, and hence remarkably enhanced mobility, than their RNA-based counterparts (Fig. 2b) — preserve nucleic acid folding more efficiently.”

2. Fig. S25 – I do not see the results for R10/E10. Why is there a reduction in the intensity of the control with time (Fig. S25B)? Is it just an experimental error, or is there some degradation? What does this mean in terms of the small increases observed for the coacervates? Why is the emission different in Fig. S25A and B? The same scale should be used for both panels.

We thank the Reviewer for noticing the oversight about R₁₀/E₁₀ (now removed from the caption).

The differences between intensities at different time points in the control are not statistically significant. The increase in fluorescence intensity observed over time in the coacervate samples, but not in solution, suggests that the partitioning of the pre-assembled aptamer is a slow process that reaches completion, in our experience, after 60 minutes. However, we agree with the Reviewer that including the two panels is confusing, as those experiments were performed separately to confirm that the variation in fluorescence emission in coacervates was not the same as that seen in solution. We have now removed panel b from Fig. S25 to avoid confusion.

3. The subtitle “Primitive coacervates enable RNA polymerization,” though correct, is misleading; polymerization is more effective in solution. The title seems to hint that the reaction occurs in the coacervates and renders some functionality to the coacervation. To my disappointment, the effect of the coacervation is to enable RNA polymerization in the dilute phase by sequestering peptides that would otherwise inhibit the reaction. While this is an interesting result, it does not support the idea that coacervates act as early cells, where one would expect some function to occur within the coacervate.

As mentioned above, in our manuscript we aimed to demonstrate the compatibility between RNA primer extension and primitive coacervates, and thus measured the efficiency of primer extension in solution. While we cannot exclude that the reaction takes also place in the coacervate phase, the extremely low volume of the coacervate phase does not allow us to separate the two phases, and thus quantify the contribution to primer extension of the condensed phase. However, we disagree that the primer extension reaction needs to occur inside coacervates to have significance.

Primer extension reactions have been frequently studied in primitive membrane-bound compartments, which drastically hinder the reactivity of the active species and reduce the efficiency of the reaction compared to the results obtained in solution. Unlike membrane-bound compartments, primitive coacervates are here shown to enable RNA polymerisation at rates comparable to solution-phase chemistry, despite the presence of inhibiting species, such as reactive peptides. As such, we believe our results demonstrate not only the inevitability of primitive coacervation in prebiotic mixtures comprising short peptides and oligonucleotides but also the compatibility of primitive coacervates as functional compartments (e.g., for oligonucleotide storage) with one of the most fundamental prebiotic RNA chemical processes proposed so far. An ongoing project in our lab aims at evaluating if non-enzymatic RNA elongation and ribozyme-catalysed polymerisation take place in the condensed or dilute phases of primitive coacervates – these findings will be the subject of a separate manuscript.

4. In Fig 4f there is a correlation between the PE efficiency and the viscosity of the coacervates as determined by FRAP, showing that the efficiency is reduced as the viscosity increases and as a conclusion the authors write “ we reason that the coacervate phase, despite its minimal volume fraction significantly contributes to the efficiency of RNA polymerization” Nevertheless, there is no explanation as to how. Is it possible that the dilute phase viscosity also increases? It is related to the amount of peptide recruited; the more, the better. The authors could have labeled the peptide and followed its concentration within the coacervates. Do they have any evidence that the reaction also occurs within the coacervates?

We apologise for the lack of clarity on this point. Untangling the effect of the viscosity of coacervates from the efficiency of primer extension is not trivial. While the viscosity of the dilute phase is not expected to significantly change between DNA- and RNA-based coacervates (due to the low concentrations of the polymers in solution), the increased partitioning of the peptide in RNA-based coacervates (Fig. 3a) might explain, at least partially, the increased primer extension efficiency within DNA-based coacervates. Moreover, the lower viscosity of DNA-based coacervates likely enables a faster recruitment of all reactants, and a faster exchange with the solution phase. We aim to definitively address this point with our future manuscript stemming from the ongoing project mentioned above on where non-enzymatic RNA elongation takes place in biphasic systems.

Minor issues

Please add units to the x-axis of Fig. 4f

Figure 4A : for R4/DNA16, the rim of the droplets shows stronger fluorescence than the center, in contrast to the other contracts. Please explain.

We have now included an explanation of this aspect, related to the higher viscosity of coacervates comprising longer DNA oligonucleotides and the slower diffusion of pre-folded RNA aptamers, which result in slower partitioning (and accumulation at the rim of the coacervates).

We modified the main text as follows:

“Interestingly, the fluorescence in R_4/DNA_{16} was not only attenuated, but also concentrated at the edge of the droplets, suggesting a slower diffusion of the aptamer within the coacervates.”

Fig. 1d - The curve for RN/HNA8 is not linear.

We apologise to the Reviewer for the lack of clarity. While a linear fit is expected for the HNA₈ system, given the linearity of the DNA₈ and RNA₈ systems, the linear fit we used for R_N/HNA₈ in Fig. 1d provides an overestimation of the minimal peptide length required for coacervation. As such, we believe the fit is not deceiving. For transparency, we have now included in the figure caption the R² values as an indication of the goodness of the fit.

Reviewer #3 (Remarks to the Author):

The study demonstrates that short, mixed-sequence oligonucleotides can form coacervates with tri- or dipeptides under conditions plausible for prebiotic chemistry, reflecting the complexity of early Earth environments. It finds that RNA-based coacervates are uniquely stable compared to those formed with DNA or peptides alone, challenging previous assumptions about the roles of DNA and peptides in early compartment formation and highlighting RNA's significance in prebiotic chemistry. Additionally, the research reveals that DNA unexpectedly enhances the diffusion of reactive oligonucleotides, which is important for non-enzymatic RNA polymerization. By systematically comparing different types of coacervates under identical conditions, this work redefines our understanding of early compartmentalization and supports a model for the co-emergence of nucleic acids and peptides, marking a significant advance in origins-of-life research.

The authors investigated RNA folding within coacervates by introducing the Broccoli aptamer as a model RNA. While this is a compelling approach to assess RNA structure in these primitive compartments, it would be valuable to discuss the degree of cooperativity between RNA folding and peptide interactions within the coacervate phase. The study shows that coacervates formed from short DNA and arginine peptides, which feature weaker and more dynamic interactions, better preserve the aptamer's folded structure compared to RNA-based coacervates, where stronger peptide-RNA interactions can partially hinder proper folding. This suggests that while coacervates can promote RNA folding by concentrating RNA and providing a unique microenvironment, the nature and strength of peptide–nucleic acid interactions are critical: excessive or overly strong interactions may reduce RNA mobility and folding efficiency, whereas more fluid, less interactive coacervates (such as those with DNA) support folding more cooperatively.

To strengthen the manuscript, I recommend a more detailed and quantitative exploration of RNA–peptide cooperativity within coacervates, including additional structural and dynamic analyses.

Further discussion on how the balance between stability and fluidity in these compartments modulates RNA–peptide cooperativity would enhance our understanding of the functional potential of prebiotic coacervates.

The Reviewer raises a very interesting point about RNA-peptide cooperativity. However, it is unclear to us whether the Reviewer refers to the cooperativity of NA-peptide coacervates and prebiotic RNA or peptide chemistry, or the cooperativity between nucleic acids, specifically RNA, and peptides in coacervation. While we agree that the first aspect is worth exploring further, and, as mentioned above, an ongoing project in the lab aims at deciphering the relationship between the coacervate phase and prebiotic RNA processes, we believe our manuscript addresses, at least partially, the second aspect. Specifically, we chose to focus on the minimal requirements for nucleic acids to undergo coacervation in the presence of model peptides. A systematic exploration of the prebiotic peptide space is instead the subject of an ongoing project in the lab. We are confident that these two studies, combined, will provide a meaningful step towards understanding the emergence of cooperativity between RNA and peptides on early Earth.

Complement fluorescence data with additional structural techniques such as NMR or the molecular simulations, to directly probe RNA conformational states and dynamics within coacervates. This could clarify whether peptides stabilize native RNA folds or induce alternative structures.

We agree with the Reviewer that exploring the ability of coacervates to preserve, enable or enhance RNA (and DNA) folding is key to understanding a variety of RNA-related processes that could have occurred on early Earth, including ribozyme-driven chemistry. Our data, depicted in Fig. 4, already suggest that primitive coacervates do not impede the formation of primer-template duplexes, which are required for efficient non-enzymatic RNA elongation. Similarly, Ref. 51 shows that DNA duplexes are maintained in Arg-based coacervates. Yet, we believe that investigating the ability of RNA to fold into functional secondary structures is beyond the scope of our manuscript, and it will be explored in future investigations that combine molecular simulations and structural data obtained by, e.g., NMR and CD spectroscopies.